# Dissociated sequential activity and stimulus encoding in the dorsomedial striatum during spatial working memory

Hessameddin Akhlaghpour[1], Joost Wiskerke[1,2], Jung Yoon Choi[1,2], Joshua P Taliaferro[1], Jennifer Au[1], Ilana B Witten[1,2]*

[1]Princeton Neuroscience Institute, Princeton University, Princeton, United States; [2]Department of Psychology, Princeton University, Princeton, United States

**Abstract** Several lines of evidence suggest that the striatum has an important role in spatial working memory. The neural dynamics in the striatum have been described in tasks with short delay periods (1–4 s), but remain largely uncharacterized for tasks with longer delay periods. We collected and analyzed single unit recordings from the dorsomedial striatum of rats performing a spatial working memory task with delays up to 10 s. We found that neurons were activated sequentially, with the sequences spanning the entire delay period. Surprisingly, this sequential activity was dissociated from stimulus encoding activity, which was present in the same neurons, but preferentially appeared towards the onset of the delay period. These observations contrast with descriptions of sequential dynamics during similar tasks in other brains areas, and clarify the contribution of the striatum to spatial working memory.

*For correspondence: iwitten@princeton.edu

**Competing interests:** The authors declare that no competing interests exist.

## Introduction

One of the most fundamental components of cognition is working memory, the ability to remember information over short periods of time and use the memory to guide ongoing behavior (*Baddeley, 1992*, *2003*; *Cowan, 2008*). Given the fundamental nature of working memory, much effort has been invested in studying its underlying neural mechanisms, and several cortical regions have emerged as important contributors. For example, neural activity in prefrontal and parietal cortex encodes stimuli during the delay period of working memory tasks, suggesting these areas contribute to maintaining the memory (*Arnsten, 2011*; *Baeg et al., 2003*; *Erlich et al., 2015*; *Funahashi et al., 1989*; *Fuster and Alexander, 1971*; *Guo et al., 2014*; *Hanks et al., 2015*; *Harvey et al., 2012*; *Horst and Laubach, 2012*; *Jung et al., 1998*; *Kojima and Goldman-Rakic, 1982*; *Lak et al., 2014*; *Powell and Redish, 2014*; *Romo et al., 1999*; *Schoenbaum and Eichenbaum, 1995*; *Shadlen and Newsome, 2001*; *Spellman et al., 2015*; *Wang et al., 2013b*; *Yoon et al., 2008*).

However, working memory is not implemented merely in cortex, but instead emerges from the interaction between cortical and subcortical areas (*Floresco et al., 1997*; *Kopec et al., 2015*; *Parnaudeau et al., 2013*), with the striatum as a key subcortical region. For example, human imaging studies have noted increased activation of the striatum during working memory tasks (*Chang et al., 2007*; *Lewis et al., 2004*; *Olesen et al., 2004*; *Postle and D'Esposito, 1999*). In addition, in the primate caudate, metabolic activity and single cell recordings point to elevated activity during spatial working memory (i.e. tasks that involve a memory for location) (*Kermadi and Joseph, 1995*; *Levy et al., 1997*). Finally, electric stimulation or lesions of the primate caudate, as well as pharmacological silencing of the analogous region in rats, the dorsomedial striatum (DMS), leads to disruptions of spatial working memory (*Balleine and O'Doherty, 2010*; *Cohen, 1972*; *Mordvinov, 1981*; *Rosvold and Delgado, 1956*; *Spencer et al., 2012*; *Stamm, 1969*).

Neural correlates of working memory in the striatum have been characterized in the case of relatively short delay period (1–4 s), but less is known about striatal dynamics in the case of longer delay periods (*Antzoulatos and Miller, 2011*; *Chiba et al., 2015*; *Hikosaka et al., 1989*; *Histed et al., 2009*; *Kawagoe et al., 1998*; *Kermadi and Joseph, 1995*; *Pasupathy and Miller, 2005*). This is a significant knowledge gap, given that animals (and humans) can remember stimuli over many seconds in real world situations.

We sought to answer several questions in this study. First, is sustained delay-period activity a feature of striatal activity in the case of long delay periods (>4 s), as has been observed in primates for short delay periods (1–4 s)? (*Hikosaka et al., 1989*; *Kawagoe et al., 1998*; *Schultz and Romo, 1988*; *Schultz et al., 1994*) If so, does that activity encode the memory of the stimulus throughout the delay period?

Another possibility is that there is sequential transient activation of neurons in the striatum during the delay period. This is a reasonable hypothesis, given that (1) memory-encoding sequences have been observed in cortical and hippocampal areas in working memory tasks (*Fujisawa et al., 2008*; *Harvey et al., 2012*; *Horst and Laubach, 2012*; *MacDonald et al., 2013*; *Pastalkova et al., 2008*), and (2) the striatum is known to exhibit sequential activity in tasks that do not directly involve working memory (*Lustig et al., 2005*; *Matell and Meck, 2000*, *2004*; *Mello et al., 2015*). If we do observe sequences during a working memory task, do those sequences encode the memory of the stimulus throughout the delay period, as has been observed in other brain regions?

To address these questions and characterize striatal dynamics during working memory, we trained rats to perform a spatial working memory task that involved long delay periods (up to 10 s). We recorded single unit activity during this task from the DMS, a region involved in spatial working memory and other related aspects of cognition (*Corbit and Janak, 2007*; *Jin et al., 2014*; *Kimchi and Laubach, 2009*; *Ragozzino et al., 2002*; *Stalnaker et al., 2010*; *Wang et al., 2013a*; *Yin et al., 2005*). We used information theoretic analyses and population decoding to characterize the neural dynamics in the recorded population.

## Results

### Behavioral performance

Rats were trained on an operant delayed non-match to position (DNMP) task (*Figure 1A,B*) (*Dunnett et al., 1988*). At the beginning of each trial, a sample lever would appear in one of two possible locations on the front wall of the chamber. The rat was required to press the sample lever, at which point the lever would retract into the wall and the delay period began. The end of the delay period was signaled by the illumination of the nose port on the back wall of the chamber (delay period duration of 1 s, 5 s, or 10 s, determined randomly). During the delay period, the nose port remained inactive (i.e. the light in the nose port remained off and nose-poking has no effect), and the rats spent the majority of their time at the nose port (*Figure 1B*, bottom-center panel; *Figure 1—figure supplement 1A*), waiting for the variable length delay period to terminate. After the illumination of the nose port, the rat was then required to place its nose into the illuminated port in order for both levers to appear in the front wall of the chamber. If the rat subsequently pressed the choice lever that did not match the initial sample lever ('non-match'), it would receive a liquid reward in a central receptacle (*Figure 1A–B*).

Of note, the rats' accuracy in this task declined with the length of the delay period ($p<10^{-4}$ repeated measures ANOVA, *Figure 1C*). This delay-dependence provides validation that the short-term memory component of the task played a role in the rats' performance, as expected in a working memory task.

In addition, omission rates and response latencies during the task revealed that the rats were highly engaged during the course of each trial. Omission rates were low regardless of the length of the delay period (<10%, *Figure 1—figure supplement 1B*). In addition, median response latencies were short for all subjects for all 3 types of response (*Figure 1D*; median sample press latency < 3.5 s, median nose-poke latency < 1 s, median choice press latency < 3.5 s).

To determine if neural activity in the DMS contributed to performance of the DNMTP task, we inactivated the DMS using the GABA$_A$ agonist muscimol, while rats performed the task (*Figure 2A*). We found a significant decline in choice accuracy with the infusion of muscimol in comparison to the

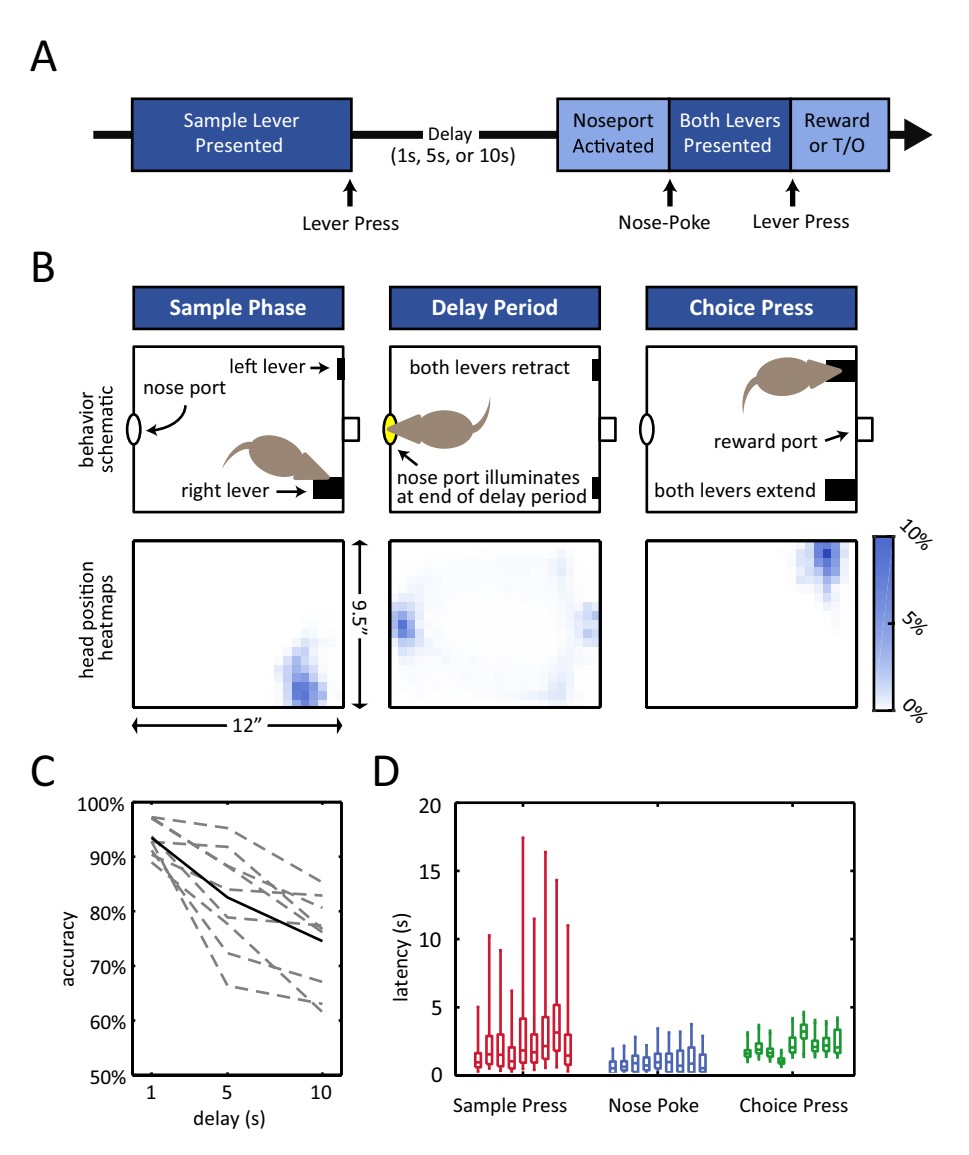

**Figure 1.** Delayed non-match to position (DNMP) task. (**A**) Schematic illustration of the task structure. A trial starts with the presentation of the sample lever at one of two locations ('sample press'). By pressing the sample lever, the rat initiates the delay period (1 s, 5 s, or 10 s duration). At the end of the delay period, the nose port on the back wall of the chamber is activated, and by entering the port (i.e. nose-poking), both levers are presented. By pressing the lever that does *not* correspond to the sample lever at the beginning of the trial (i.e. choice press), the rat receives a reward; pressing the other lever results in a timeout (T/O). (**B**) Top: Schematic illustration of the chamber and the rat's position at the time of the sample lever press (left), the delay period (middle), and the choice lever press (right). Bottom: Occupancy-map of the rats' head position for right-sample trials at the time of the sample lever press (time window from 250 ms before to 250 ms after lever press), delay period (center, time window is the entire delay period for 10 s delay period trials only), and choice lever press (right, time window from 250 ms before to 250 ms after lever press). Occupancy-maps were generated by averaging the occupancy-maps of the 9 rats used in the electrophysiology experiments. Individual occupancy-maps were calculated by binning the head positions over the respective time windows into 0.5" × 0.5" tiles (covering the 9.5" ×12" chamber). (**C**) Accuracy is delay dependent, decreasing as the duration of the delay period increases (p<10$^{-4}$). Solid black line represents mean accuracy, and dotted grey lines represent the accuracy for each individual rat. Accuracy was calculated from the final recording session. (**D**) Response latency for each individual rat for sample press (time from sample presentation to sample press), nose-poke (time from nose port activation to nose-poke), and choice press (time from choice presentation to choice press). Center of each boxplot represents the median, edges correspond to 25th and 75th percentile, and whiskers correspond to fifth and 95th percentile.

*Figure 1 continued on next page*

*Figure 1 continued*
The following figure supplement is available for figure 1:

**Figure supplement 1.** Behavior during the delayed non-match to sample task.

infusion of saline (*Figure 2B–C*: p<0.001 effect for infusion day, repeated measures ANOVA, n = 7). Importantly, muscimol infusion had no significant effect on median response latencies for rats, suggesting minimal motor impairment (*Figure 2D*). We also examined the effects of muscimol on the number of trials performed, sample omission rates, trial abort rates, sample bias, and choice bias and found that none of these measures were significantly affected by DMS inactivation across the population (*Figure 2—figure supplement 1*, Wilcoxon signed rank test, n = 7). However, a subset of rats did perform fewer trials when treated with muscimol, which is consistent with the DMS being implicated in motivation or response vigor (*Wang et al., 2013a*).

## Sequential activation of neurons

Spiking activity for 105 neurons was isolated from the DMS of 9 rats (summary of electrode localizations in *Figure 3A*). Between 5 and 25 units were recorded in each subject (*Figure 3B*, inset). The average firing rates of these units tended to be low (population mean < 6 Hz) (*Figure 3B*), consistent with previous reports of medium spiny neurons in the dorsal striatum (*Berke, 2008*; *Berke et al., 2004*; *Mallet et al., 2005*).

To determine if and how the neurons' firing rates evolved during the course of a trial, we examined the time-varying firing rate of each neuron relative to the onset of the delay period. The recorded units displayed a diversity of firing rate patterns; for instance, the example unit depicted in *Figure 4A* showed a transient peak in its firing rate a few seconds into the delay period. In order to visualize the activity across the neural population, we ordered the units by the time of the peak firing rate for the 10 s delay trials, and then generated firing rate heat-maps for 1 s, 5 s, and 10 s delay-

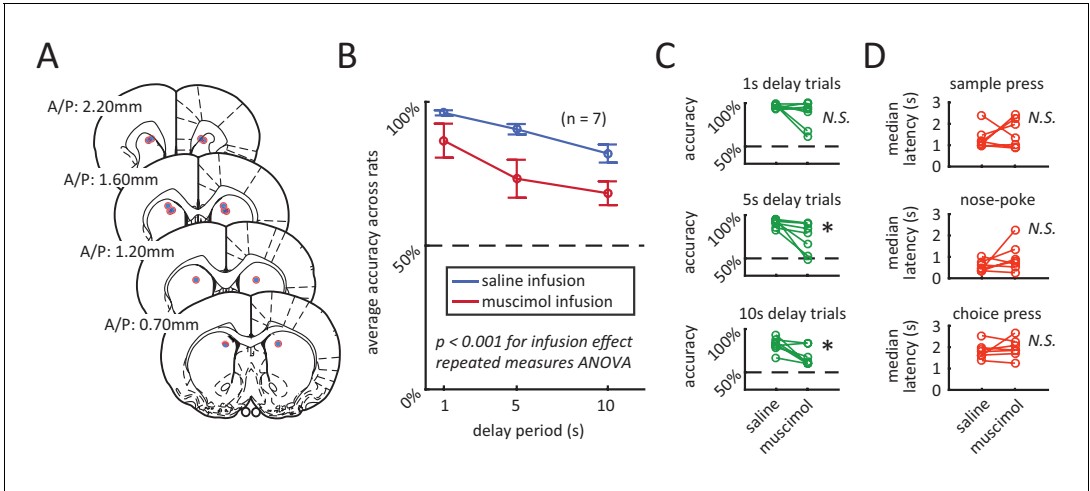

**Figure 2.** Inactivation of dorsomedial striatum (DMS) impairs accuracy during the DNMP task. (**A**) Red circles with blue fillings represent location of injection cannula tips as revealed by post-mortem histological analysis. (**B**) Muscimol infusion in DMS impairs accuracy when compared to infusion of saline (p<0.001; effect of muscimol repeated measures ANOVA, no significant interaction effect between delay length and infusion day). Error bars are ± 1 SEM across 7 rats. (**C**) Accuracy was significantly impaired for the 5 s and 10 s delay trials (*p<0.05; Wilcoxon signed rank test) but not for 1 s delay trials (p=0.38; Wilcoxon signed rank test, n=7 rats). (**D**) No signficant effect of DMS inactivation on median response latency for sample press (top, p=0.58), nose-poke (middle, p=0.28), and choice press (bottom, p=0.92; Wilcoxon signed rank test, n=7 rats).
The following figure supplement is available for figure 2:

**Figure supplement 1.** Additional behavioral measures were not signficantly effected by muscimol treatment.

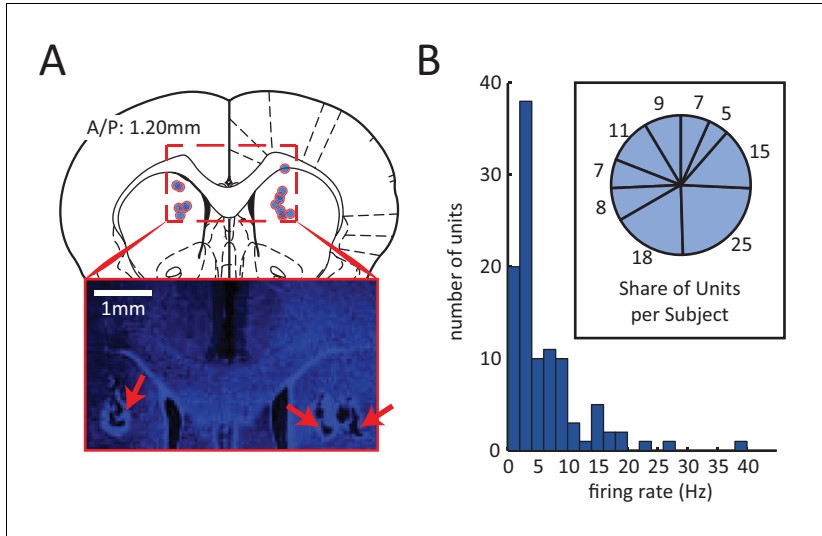

**Figure 3.** Electrophysiological recording of neural activity in the dorsomedial striatum (DMS). (**A**) Red circles with blue fillings correspond to location of DMS electrode array tips in 9 rats. Electrodes array rows were oriented in the A/P direction to span a length of 1.6 mm, centered at A/P: 1.2 mm. Inset: Example image of electrode lesions in DMS (blue DAPI stain; 1 mm scale bar). (**B**) Histogram of mean firing rate of 105 isolated units. Inset: number of units isolated from each rat.

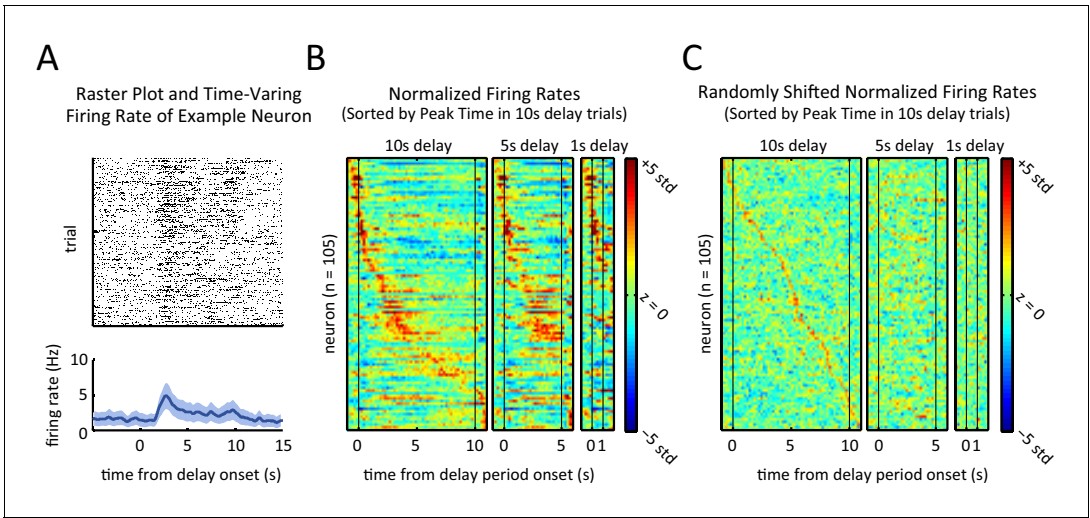

**Figure 4.** Sequential, transient peaks in the firing rate span the delay period. (**A**) Raster plot (top) and firing rate (bottom) for an example neuron (aligned to the onset of the delay period). Shaded area represents ± 1 SEM. (**B**) Heat-maps represent the z-scored firing rates for all units for the 10 s, 5 s, and 1 s delays, aligned to the onset of the delay period. Each row is a single unit. Rows in all three plots were sorted by the peak firing rate time in the 10 s delay trials (left-most plot). (**C**) Same as **A**, but spike times were randomly shifted relative to behavioral timestamps.

The following figure supplements are available for figure 4:

**Figure supplement 1.** Presence of sequential activity when controlling for position, and when aligning to event markers other than delay period onset.

**Figure supplement 2.** Ridge-to-background analysis to quantify the presence of firing rate sequences.

duration trials (*Figure 4B*). Neurons displayed sequential peaks of activity that spanned the entirety of the 10 s long delay period (*Figure 4B*, left panel). The firing rates for the shorter delay trials displayed similar sequential activity throughout the delay period (*Figure 4B*, middle and right panel), when ordered based on the peak times for the 10 s delay-duration trials (p<10⁻⁷, Spearman correlation test between peak times of 5 s-delay trials and 10 s-delay trials). Note, by design, only neurons that were engaged during the first 1 s or 5 s of the 10 s-long delay period could be part by the sequence for the shorter delay trials, which is why the sequences involved fewer neurons for the shorter delay periods.

To further validate and characterize the finding of sequential firing rate peaks, we repeated the same analysis after randomly shifting the time of the recorded spikes and the behavioral timestamps for each neuron (*Figure 4C*). The shifted data (*Figure 4C*) differed substantially from the non-shifted data (*Figure 4B*), confirming that the activity sequences that we observed were not an artifact of ordering the activity based on the peak response. In the shifted data, sequences were non-existent in the 5 s and 1 s duration trials and there was no correlation between the order of the peak firing rate for the 10 s delay-duration trials and for the 5 s delay-duration trials (p=0.29, Spearman correlation test). As another method to quantify the presence of sequential firing rate activity, we calculated the ridge-to-background ratio (*Harvey et al., 2012*; see Materials and methods). Ridge to-background ratios were significantly greater than chance level for all delay period lengths (*Figure 4—figure supplement 1A*, p<0.001; one tailed test using the ratios from the randomly shifted data as the null distribution).

In comparison to the shifted data, the distribution of peak firing rate times in the real data revealed a bias towards the onset of the delay period (*Figure 4—figure supplement 2E*). In other words, although the sequences spanned the delay period, there were more peaks towards the beginning of the delay period relative to what would be expected by chance (p<0.01, Wilcoxon rank sum test).

Because the rats were not stationary during the entirety of the delay period, the possibility arises that the sequences were in fact a byproduct of neural selectivity for position or movement. To control for this possibility, we re-calculated the sequences while using only the subset of time during which the rats were positioned in front of the nose port (*Figure 4—figure supplement 2A–D*). We found that even when controlling for the rats' position by excluding times that they were not at the nose port, sequences were evident that closely matched the sequences based on the full dataset (*Figure 4—figure supplement 2B*). Ridge-to-background analysis confirmed that the presence of firing rate sequences was statistically significant, even when using only data from times when the rat was at the nose port (*Figure 4—figure supplement 1B*). We therefore conclude that sequential activations of DMS neurons are not a by-product of responses to movement or position.

As an additional approach to control for the possibility that sequential firing activity might be due to position and locomotion selectivity, we found that time in the delay period is a significant predictor of neural activity, even when taking into account position/locomotion variables as alternative predictors. More specifically, we modeled each neuron's spiking activity using two generalized linear models (GLMs), one with position related predictors only (i.e. head position, head direction, and velocity – all calculated using head tracking) and the other with both the position-related predictors as well as time from the delay period onset as a predictor (see Materials and methods for details). Time from delay onset significantly improves the model in 80% of the neurons (84/105) in comparison to a model with only the position/locomotor variables (p<10⁻⁴, likelihood ratio test comparing models with and without time from delay period as predictor). This demonstrates that the time-dependent changes in firing rates during the delay period cannot be accounted for by position or the other variables we tested. Note that sequences were not specific to the delay period, as we observed similar sequential activation of neurons when time-locking to other task events (*Figure 4—figure supplement 2F*, *Figure 4—figure supplement 1C*).

## Transient encoding of the sample stimulus

To determine how firing rate sequences during the delay period relates to the sample lever selectivity of the neurons, we compared neural activity for trials when the sample lever was in the right versus the left location, excluding error trials and omission trial. Individual neurons exhibited different patterns of sample selectivity (*Figure 5A*). One example neuron displayed no choice-selective modulation (left panels, *Figure 5A*), another neuron was only activated during right sample trials (middle

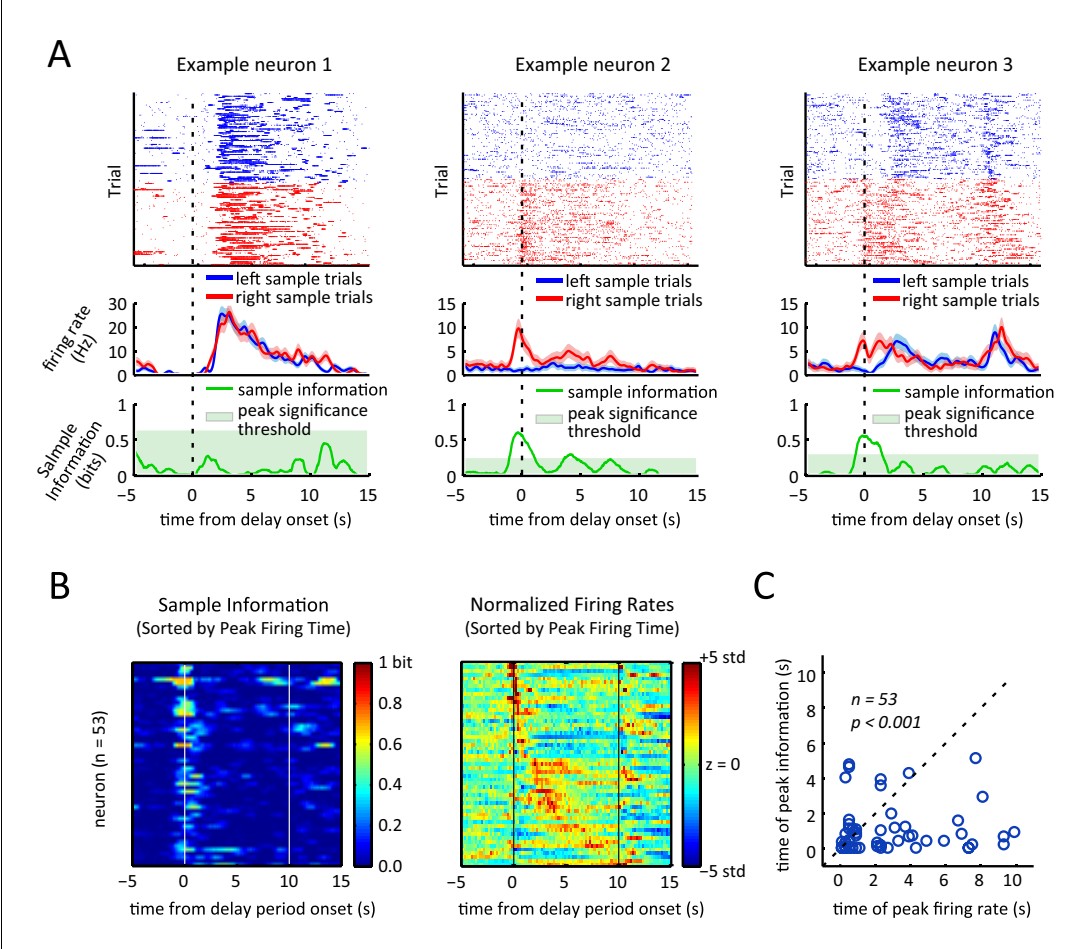

**Figure 5.** Sample lever information peaks are clustered towards the beginning of the delay period and are not correlated with firing rate peak times. (A) Each column represents data from an example neuron, with neural activity aligned to the onset of the delay period, and activity color-coded in red for right sample trials and left for blue sample trials. Top: raster plot, with each dot representing a spike. Middle: firing rates for left and right sample trials. Shaded area represents ± 1 SEM. Bottom: Information about sample as a function of time. Shaded area represents the 99th percentile of the maximum information expected by chance across the entire time interval, calculated by shuffling the sample labels for the trials. (B) Left: Heat-map of sample information as a function of time for the 53 neurons that had significant information peaks within the 10 s delay period (p<0.01; one-tailed test using shuffled data for null distribution of peak information within the 10 s delay period). Right: Heat-map showing z-scored firing rates of the same neurons depicted in the left plot. Neurons in both the left and right panels are sorted by the peak firing rate in the delay period, such that a neuron on the left plot appears in the same row in the right plot. (C) Time of peak sample information plotted against the time of peak firing rate for 53 sample encoding neurons (where 0 corresponds to the onset of the delay period). The data reveals sample information peaks times occur earlier than firing rate peak times (p<0.001; Wilcoxon signed rank test), and a lack of correlation between the time of peak firing rate and time of peak sample information (r=0.06, p=0.69; Pearson correlation test). All panels are calculated using 10 s-delay correct trials only.

panels, *Figure 5A*), and finally a neuron displayed different patterns of activation for left and right sample lever trials (right panels, *Figure 5A*).

Given the diversity of responses in individual neurons, we turned to mutual information as a method to quantify the amount of information about the sample stimulus encoded in the firing rate as a function of time (*Borst and Theunissen, 1999*; *Dayan and Abbott, 2001*; *Rieke et al., 1999*). We calculated the time-varying mutual information between the sample stimulus and the spike trains of individual neurons (within a 0.5 s sliding window), by assuming that spiking of each neuron is a Poisson point process with a time varying rate determined by the sample stimulus and time from the delay onset (see Materials and methods for details). We found many neurons showed transient information peaks (e.g. example neurons 2 and 3 in *Figure 5A*). To determine if these peaks were statistically significant, we calculated the distribution of the maximum information across the entire delay

period that would be expected by chance for each neuron by repeatedly shuffling the labels specifying whether a trial had a left or a right sample and recalculating the mutual information. A neuron was considered to significantly encode the sample stimulus during the delay period if its peak sample information was larger than the 99th percentile of that distribution (green shaded area in the bottom panels of *Figure 5A*). By this criteria, 53 of the 105 neurons showed significant encoding of the sample stimulus at some point during the delay period (p<0.01) with peak information values ranging between 0.07 to 0.69 bits (average peak information was 0.34 bits).

To compare firing rate sequences to stimulus encoding across the recorded population, we compared the time-varying firing rate alongside the sample information for the subset of neurons that encoded the stimulus significantly at some point in the delay period (*Figure 5B*). Surprisingly, the sample information peaks did not display a sequential organization to match the peak firing rate sequence. In fact, the time of the peak sample information did not correlate significantly with the peak firing rate time (*Figure 5C*, Pearson correlation test; r=0.06, p=0.69). In fact, sample information most often peaked towards the onset of the delay period, with more than 80% of the stimulus encoding neurons (45/53) displaying peak information within the first 2 s of the delay period. On average, peak sample information was significantly earlier than peak activity (p<0.001; Wilcoxon signed rank test. Only neurons with significant information peaks were considered for this analysis).

We found that individual neurons encode the sample stimulus at specific times in the delay period, and that for many neurons sample encoding was strongest towards the beginning of the delay period (*Figure 5*). To quantify how sample encoding in the DMS changes throughout the course of the trial we constructed a population decoder that takes spiking activity of all the recorded neurons within a 500 ms sliding window and evaluated its performance using leave-one-out cross validation (*Brown et al., 2004*; *Gerwinn et al., 2009*; *Ma et al., 2006*; *Pouget et al., 2003*; *Quian Quiroga and Panzeri, 2009*). (In order to combine neurons across rats, we create pseudo-trials; see Materials and methods). This provided us with a time varying sample decoding accuracy representing how well DMS neurons encode the sample stimulus at different parts of the delay period (*Figure 6*). In agreement with our findings from *Figure 5*, sample decoding was highest at the onset of the delay period (at the time of sample lever press) and declined over the course of the delay period (*Figure 6A*).

In order to compare the neural code in DMS for correct and error trials, we calculated the output of the decoder on error trials (after the decoder was trained on correct trials). This allowed us to determine when in the trial neural activity encoded the identity of the sample versus the identity of the choice. This distinction is difficult to make without analysis of error trials, because in correct trials the sample and choice levers perfectly predict one another. Given that the decoder for error trials is trained on correct trials only, a significantly lower than chance level decoding accuracy for error trials means higher than chance level encoding of the choice lever. The accuracy of decoding error trials was high towards the beginning of the delay period, quickly dropped and stayed at chance level throughout most of the delay period, and fell below chance near the choice lever press time (*Figure 6B*). This means that neural activity around the time of the sample lever press encodes the sample lever identity, as opposed to the rat's future choice. In contrast, at the time of the choice lever press, neural activity encoded the identity of the choice lever, as opposed to the sample identity.

Given that rats are allowed to move freely during the delay period, we considered the possibility that sample encoding early in the delay period (*Figure 5B* and *Figure 6*) may simply be a result of encoding of different positions/motor actions across left and right sample trials. To address this, we examined a time point during the delay period during which the movement trajectories converge for left and right sample trials. Following the sample press, rats typically move to the opposite wall and wait in front of the nose port for the delay period to end (*Figure 1—figure supplement 1A*). Instead of aligning trials to the beginning of the delay period (i.e. the sample lever press), we aligned trials to the moment the rats first arrived at the nose port, and excluded the small subset of trials in which they left the nose port within 1.5 s of the nose port arrival. We show that head position is very similar across left and right sample trials following nose port arrival (*Figure 6—figure supplement 1A*), yet our population decoder was able to decode the sample stimulus with high accuracy, even during the first 1.5 s when the rat strictly maintains its head position in front of the nose port (*Figure 6—figure supplement 1B*). This is important as it provides evidence of memory encoding within striatal neurons during the delay period while controlling for the rats' position on left versus right sample trials.

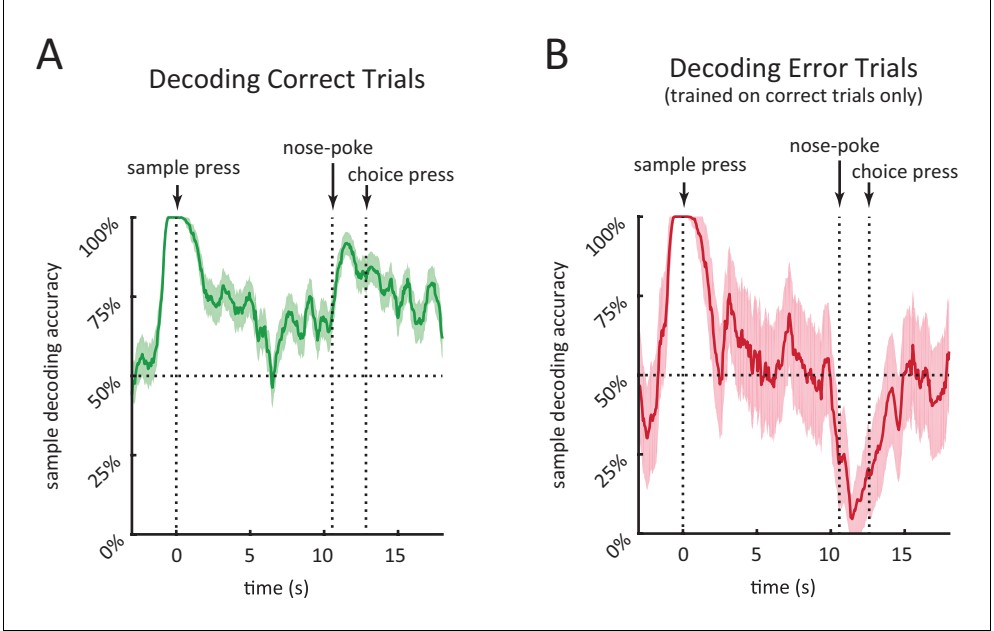

**Figure 6.** Performance of population decoder for sample lever identity. (**A**) Accuracy of decoding sample lever from population spiking data within a 0.5 s-wide sliding window in 10 s-delay correct trials. Decoder was trained on 10 s-delay correct trials and evaluated using leave-one-out cross validation. (**B**) Same as A but decoding 10 s-delay error trials. Decoder was trained on all 10 s-delay correct trials. In both panels, the dotted lines marked with 'nose-poke' marks the median time of nose-poke, and the dotted line marked as 'choice press' marks the median time of choice press. Shaded area represents ± 1 SEM.

The following figure supplement is available for figure 6:

**Figure supplement 1.** Encoding of sample lever when controling for head position.

In an alternative approach to determine if the sample memory is present in DMS activity in a form beyond encoding of position/locomotion, we sought to test how well the sample stimulus can improve predictions of firing patterns when including other variables as predictors. Towards this end, we modeled each neuron's spiking activity using two generalized linear models (GLMs), one with position/locomotion related predictors only and the other with both task-related and position/locomotion related predictors. The position/locomotion predictors we used were head position, head direction, and head velocity and the task-related variables were time from delay period onset, sample stimulus of current trial, and the interaction between time and sample stimulus. We performed a nested model likelihood ratio test for each neuron to compare the two models, to determine if the addition of the sample memory related variables improved the model (see Materials and methods for details). For more than 90% of neurons (95/105), the model significantly improved when including the sample memory related variables (p<0.0001; Pearson's chi-squared test). These neurons included 51 of the 53 sample encoding neurons identified through the mutual information analysis (*Figure 5B–C*).

As a final approach to address position differences between left and right sample trials, we considered the possibility that differences between the left-sample and right-sample spiking patterns may be due to the fact that the rat is simply in different locations or performing different actions in the two conditions. In order to determine if neurons are encoding the identity of the sample lever rather than motor actions or location in the chamber, we took advantage of the task design in that the rats performed the same lever pressing action at the same location but in different contexts of the task, i.e. either in the sample phase or the choice phase. More specifically, if striatal neurons only encode the action or the location, we would not to expect to see differences in the neural coding for sample lever presses and choice lever presses. Note that the rats' head position is very similar at the time of sample and choice lever presses (*Figure 7A*).

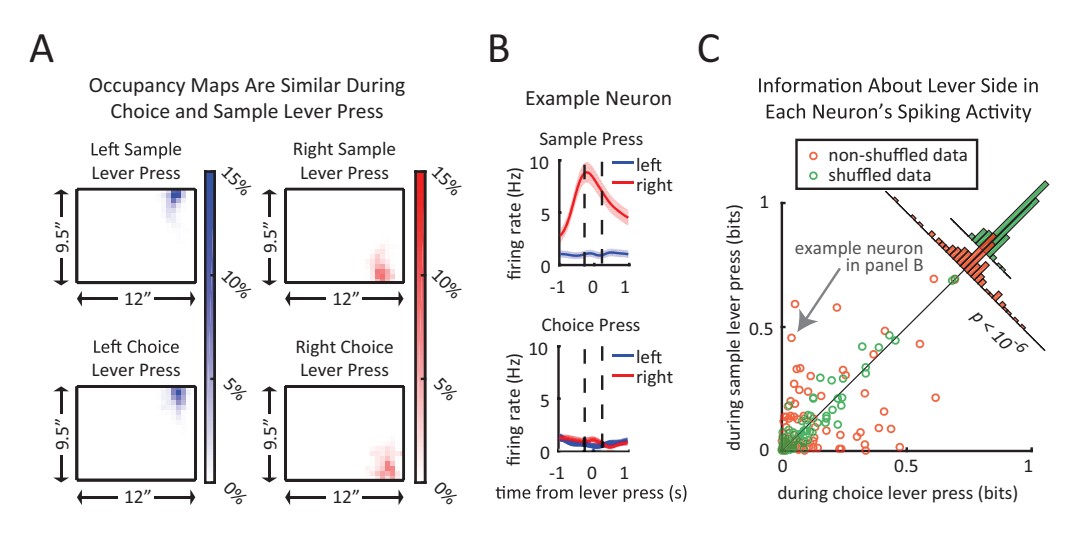

**Figure 7.** Context-dependent neural encoding of the lever press. (A) Left plots show similarity of head position during left lever press for sample lever press (top) and choice lever press (bottom). Right plots are same as left plots except for right lever press. Average head position occupancy-maps from 250 ms before to 250 ms after the lever press. Occupancy-maps were generated by averaging individual occupancy-maps of all 9 rats (0.5″×0.5″ tiles covering the 9.5″×12″ chamber). (B) Firing rates of an example neuron during sample lever press and choice lever press. This neuron displays an obvious difference in its encoding of lever side between the sample and choice presses. (C) Neurons differentiate between sample and choice presses in the amount of information conveyed about the lever identity. Each neuron's information about lever side was calculated by taking the mutual information between lever identity and spiking patterns within a 500 ms time window centered at the lever press (to match time window used in panel A). Orange circles represent sample lever information plotted against choice lever information. Green circles are obtained by recalculating the information after shuffling sample and choice labels for lever presses. Insets are histograms of choice information minus sample information for non-shuffled data (orange) and shuffled data (green). Neurons showed significantly greater difference between sample and choice information than expected by chance ($p<10^{-6}$; two sample F-test for equality of variance for (*sample information – choice information*) in shuffled versus non-shuffled data).

We found that many neurons differentiate between sample and choice presses in their encoding of lever identity. For instance, the example neuron in *Figure 7B* responds to right sample presses but not right choice presses. Such a dramatic difference cannot be explained merely as a result of encoding of motor actions, location, or body position since the rat performs the same action at a similar location but we observe very different neural responses. Much like the example neuron in *Figure 7B*, across the population of recorded neurons, some neurons preferentially encoded lever identity during the sample press while others encoded the lever identity during the choice press (*Figure 7C*). There was significantly more dispersion in the difference between sample and choice information than expected by chance (inset in *Figure 7C*, $p<10^{-5}$; two sample F-test for equality of variance for (*sample information – choice information*) in shuffled versus non-shuffled data). This analysis supports the conclusion that DMS firing patterns are not merely reflecting position or motor actions.

## Discussion

We characterized neural dynamics in the DMS during spatial working memory and found that neurons were sequentially activated throughout the course of the delay period. The sequential activity was dissociated from memory-encoding activity, which was present in the same neurons, but tended to occur earlier in the delay period. In other words, the time that a neuron most strongly encoded the sample stimulus during the delay period was not predictive of the time that it was most active.

### Insights into the striatal dynamics underlying working memory

We employed two different methods to quantify the encoding of the sample stimulus during the delay period. The first method was to quantify the mutual information between the sample stimulus

and the firing patterns for each individual neuron (*Figure 5B*), which revealed strong sample stimulus encoding at the onset of the delay period and less encoding during the delay period. The second method (*Figure 6A*) was to measure our ability to decode the sample stimulus from the entire neural population as a function of time during the delay period. The second method revealed perfect decoding accuracy at the onset of the delay period and greater than chance level accuracy during most of the delay period. Both approaches demonstrated that the sample stimulus is encoded strongly at the onset of the delay period and that stimulus encoding declines noticeably with either measure by 2 s into the delay period. Stronger stimulus encoding toward the onset of the delay period has been reported in studies of primate cortical regions, although those studies did not compare sample stimulus encoding to sequential firing activity (*Brody et al., 2003*; *Bruce et al., 1985*; *Clark et al., 2012*; *Colby et al., 1996*; *Gregoriou et al., 2012*; *Romo et al., 1999*).

Our experiments differed from most previous work in that we employed relatively long delay periods (up to 10 s). Considering that sample information in the DMS decreased substantially within several seconds (*Figure 5B* and *Figure 6A*), the transient nature of stimulus encoding would have been less obvious if our task had only employed substantially shorter delay periods. Instead, the activity would have been better described as sustained memory-encoding activity. Indeed, previous recordings in primates have reported sustained delay period activity that encodes short-term memory in a subset of caudate neurons with delay periods of 2–4 s (*Hikosaka et al., 1989*; *Kawagoe et al., 1998*; *Schultz and Romo, 1988*; *Schultz et al., 1994*). In our study, we observed some neurons with stimulus encoding during the delay period that persisted for several seconds, in agreement with prior work. However, no single neuron was found to encode the sample stimulus for the entire duration of the delay period (*Figure 5B*).

A strength of our task design is that it allowed us to compare the same motor action (i.e. lever pressing) in different contexts within the trial (*Figure 7*). This revealed that striatal neurons differentially encode the sample press and the choice press, and that encoding of the same action is dependent on the context of the action within the task. This supports the idea that the striatum is involved in encoding memories of task-related variables, and not merely encoding the rats' position or movement during the task.

Unlike most prior work performed in head-fixed primates, our rats were freely moving. Thus, we performed several analyses to control for the possibility that our conclusions of sequential firing rate activity or transient stimulus encoding was a result of motor or position confounds. This included analyzing data from times that the rat was in the nose port (*Figure 4—figure supplement 2A–D* and *Figure 6—figure supplement 1*), as well as showing that task-relevant variables remained a significant predictor of neural activity in the vast majority of neurons, even when accounting for movement or position with a generalized linear model (GLM). However, motor and position confounds may have been even more strongly excluded had we instead used a task design that restricts movement of the rat during the delay period. Additionally, in our task design the correct choice of a trial can be determined from the onset of the delay period, making it difficult to distinguish encoding of the memory of the sample stimulus from encoding of future motor actions ('retrospective' versus 'prospective' memory). A more sophisticated task design that enabled dissociation of stimulus encoding and pre-motor activity could have allowed us to make even stronger conclusions about memory encoding.

## Relevance to theoretical models of the striatal contribution to working memory

Different theories have been proposed to explain how the striatum contributes to working memory. For example, it has been suggested that the striatum may be involved in initiating the storage of new memories in cortical networks, an idea often referred to as 'gating' (*Baier et al., 2010*; *Frank et al., 2001*; *Gruber et al., 2006*; *O'Reilly and Frank, 2006*; *Braver and Cohen 2000*). According to this idea, the striatum plays a role in selecting cortical memory buffers that are to be updated and determines the time memory updating occurs, similar to the role the striatum is thought to play in initiating and selecting motor actions (*Bailey and Mair, 2006*; *Bhutani et al., 2013*; *Grillner et al., 2005*; *Kropotov and Etlinger, 1999*; *Mink, 1996*). The gating model ascribes a crucial role to the striatum at the onset of the delay period, when new information must be stored in short-term memory buffers (*Baier et al., 2010*; *Frank et al., 2001*; *Gruber et al., 2006*; *O'Reilly and Frank, 2006*; *Braver and Cohen 2000*). Interestingly, our data emphasizes the

importance of the striatum at the beginning of the delay period, as the stimulus is most strongly encoded during that period.

However, another possibility is that DMS is contributing to maintaining the short-term memory. Although the accuracy of our population decoder decreases substantially during the course of the delay period, the decoding accuracy near the end of the delay period is roughly comparable to behavioral performance even at the 10 s delays (*Figure 1C* and *Figure 6A*), indicating that activity in the DMS might be sufficient to support the animal's behavior. As another possibility, the DMS could assist cortical areas in maintaining the memory throughout the delay period, irrespective of holding the content of the memory. Memory encoding activity in cortex often appears as persistent elevated activity or sequential activations of neurons (*Baeg et al., 2003*; *Cromer et al., 2011*; *Fujisawa et al., 2008*; *Fuster and Alexander, 1971*; *Hanks et al., 2015*; *Harvey et al., 2012*; *Horst and Laubach, 2012*; *Kojima and Goldman-Rakic, 1982*; *MacDonald et al., 2013*; *Pastalkova et al., 2008*; *Powell and Redish, 2014*; *Schoenbaum and Eichenbaum, 1995*), which may be implemented using positive feedback through distributed cortico-striato-thalamic loops (*Alexander et al., 1986*; *Houk and Wise, 1995*; *Middleton and Strick, 2000*, *2002*; *Wang, 2001*). In this view, the sequential activity that spans the delay period in DMS could play a role in exciting cortical memory buffers and enabling memory encoding activity there. As yet another possibility, the stimulus memory may be stored in the striatum in a form that would not be fully detectable in the spiking activity, perhaps in short-term synaptic plasticity, as suggested by recent theoretical models (*Lundqvist et al., 2010*; *Mongillo et al., 2008*; *Stokes, 2015*). Future experiments employing transient perturbations of the striatum at sub-trial temporal precision will help address the question of whether the DMS is more important for initiating versus maintaining memories.

## Relationship to neural dynamics reported in other tasks

Sequential neural activity dynamics have been reported in the striatum in tasks that do not involve spatial working memory (*Lustig et al., 2005*; *Matell and Meck, 2000*, *2004*; *Mello et al., 2015*). These studies, taken together with our finding that firing rate sequences we observed here were dissociated from stimulus encoding activity (*Figure 5B,C*), suggests that the delay-period spanning sequences in DMS support a general role for the striatum in time keeping, rather than a specific role in working memory.

On a related note, the neural dynamics we observed in the DMS underlying spatial working memory provide an interesting contrast with previous observations of sequential activity dynamics in other regions during spatial working memory. In particular, cortical (*Astrand et al., 2015*; *Cromer et al., 2011*; *Crowe et al., 2010*; *Fujisawa et al., 2008*; *Harvey et al., 2012*; *Horst and Laubach, 2012*) and hippocampal (*MacDonald et al., 2013*; *Pastalkova et al., 2008*) areas display stimulus encoding sequences that span the delay period. This suggests a different role for the striatum in working memory relative to these other regions. Specifically, cortex and hippocampus may be involved in maintaining the memory throughout the delay period, whereas the striatum may be involved in initiating memory storage.

## Conclusions

We employed information theoretic analysis and population decoding to reveal that sequential activity and stimulus encoding are dissociated in the neural dynamics of the striatum during spatial working memory. Specifically, neurons transiently encoded the stimulus during the onset of the delay period, whereas the same neurons encoded time with sequential activity throughout the delay period.

# Materials and methods

## Behavioral task

Male Long Evans rats (7 for infusion experiments and 9 for recordings) were trained on a delayed non-match to position (DNMP) spatial working memory task in operant chambers (*Figure 1A,B*). There were two retractable levers on the operant chamber front wall (right side of schematic and heat-maps, *Figure 1B*). At the beginning of each trial, a sample lever extends out of the wall, (either on the left or the right side – randomly interleaved). The rat is then required to press that lever, after

which the lever retracts back into the wall. The rat has to remember the sample lever side for the duration of the delay period (1 s, 5 s, or 10 s – randomly interleaved). The end of the delay period is signaled by the illumination of the nose port on the back wall of the chamber (left side of schematic and heat-maps, *Figure 1B*). The rat must then poke his nose into the illuminated nose port for both levers to extend from the front wall of the chamber, and then choose one of the two levers. Following a correct lever press, i.e. a choice lever press that does not match the initial sample lever, the rat is rewarded with 40 µl of milk in the reward receptacle (*Figure 1A,B*). The rats were given 30 s to respond to the sample lever, 5 s to respond to the activated at the end of the delay period, and 5 s to respond to the choice lever. An incorrect lever press or a failure to respond within the time limit, resulted in a 5 s time-out penalty, during which the house-light is turned off. Trials were followed by 10 s long inter-trial interval (ITI). The task was designed to encourage the rats to spend the delay period at the nose port. Tracking of head-position confirmed that rats spent the majority of the delay period in the nose port (*Figure 1B* center panel and *Figure 1—figure supplement 1A*). The rats were kept on a limited food diet to motivate behavior.

## Surgical procedure

All procedures were performed in accordance with the university-approved IACUC protocol.

For the cannula implantations, 12 adult Long Evans rats (>300 g) that were previously trained in the DNMP task were deeply anesthetized and a double guide cannula was implanted bilaterally above the dorsomedial striatum (DMS) (A/P: 1.2 mm, M/L: ± 1.9 mm, D/V: −5.25 mm, taking into account the 1.25 mm or 2.75 mm projections of internal cannulas used during injections).

For the electrode implantation, 9 adult Long-Evans rats (>300 g) that had previously undergone training in the DNMP task were deeply anesthetized and electrode arrays were implanted either uni-laterally or bilaterally over the DMS. The implants were omnetics based TDT micro-wire arrays, composed of 1–2 rows of 8 polymide insulated tungsten wires (50 µm diameter, with 175 µm spacing within a row and 500 µm spacing between rows). The DMS was targeted stereotactically (A/P: 1.2 mm, M/L: ± 2.0 mm, D/V: −4.0 to −5.0 mm) and electrodes were oriented so that the length of the rows went along the anterior/posterior axis. A ground screw was implanted in a posterior location on the skull and connected to the ground wire from the array.

## Muscimol infusions

After a week of recovery time post-surgery, the rats were retrained on the DNMP task until behavior re-stabilized. Before each testing session, rats were anesthetized in an induction chamber and were then moved to a nose cone to maintain anesthesia (with 2% isoflurane) for a total duration of 20–25 min. During this period the cap and dummy cannulae were removed and internal cannulae with appropriate projection length were inserted into the guide cannula. 600 nl of saline or muscimol solution was infused at a rate of 200 nl/min. The internal cannula was removed and dummy cannulae were inserted 4 min after each infusion, to allow the solution to diffuse. Rats were given 20–25 min of recovery time after the infusion and anesthesia, before being placed in the operant chamber to begin behavioral testing. To identify the muscimol concentration that was used for testing each rat, a range of muscimol infusion concentrations (37.5–75 ng) was tested on different days, and the largest concentration for which each rat performed a sufficient number of trials during a session was identified (at least 50 trials per session). Once the concentration was identified, the rats underwent two days of additional infusion sessions. On the first day they received an infusion of saline, and on the second day they received an infusion of the muscimol concentration selected as described above. Concentration and volume were similar to values used in previous studies (*Spencer et al., 2012*; *Yin et al., 2005*).

Of the 10 implanted rats, 3 rats were not included in the final data set. One rat became sick over the course of the study. One rat did not perform the task under the range of muscimol concentrations that we infused. And one rat had a brain infection that was evident in the post-mortem histology analysis.

## In vivo electrophysiological recordings

After a week of recovery time, the rats were retrained on the DNMP task while simultaneously being habituated to having wires connecting the implant arrays to the TDT 32 channel recording system

through a motorized commutator. After two weeks of retraining and habituation, each rat underwent a single recording session of 1.5–3.5 hr in, in which 200–520 trials were recorded.

## Head tracking

All head tracking data was obtained by using the TDT RV2 system, linked to a camera directly above the chamber. The RV2 system was trained to detect two LEDs mounted on top of the rat head stage (one green and one red). Linear interpolation was used to fill in the gaps for the periods when the signal was lost for either of the LEDs. To determine when a rat was 'at the nose port' (*Figure 4—figure supplements 1–2* and *Figure 6—figure supplement 1*) we first found the median head position coordinates for when the rat made nose pokes. Whenever a rat was within 2.5" of that position, that rat was considered to be 'at the nose port'. First nose port arrival time was calculated accordingly.

## Histological analysis

After the recording experiments, under deep anesthesia, electrolytic lesions were generated at each electrode tip to enable the locations of the electrode tips to be visualized. Immediately after the lesions were generated, the rats received a transcardial perfusion of PBS followed immediately by perfusion of 4% PFA in PBS. Brains were place in 4% PFA in PBS solution for 24 hr. Then they were transferred to a solution of 30% sucrose in PBS. 40um thick sections were generated with a microtome. The relevant DMS slices were stained with DAPI and the lesions were visualized with a stereomicroscope (Leica).

## Data analysis

Spikes were detected online with amplitude thresholding. Clustering was performed manually using Plexon Offline Sorter. Any cluster that was clearly distinct from the noise cluster with reasonable spike amplitude/waveform and inter-spike interval histograms was identified as a single unit.

The firing rate heat-maps (*Figure 4B*, *Figure 5B*, and *Figure 4—figure supplement 2*) were generated by fist binning spike times relative to the specified event marker and calculating average firing rate in each time bin (calculated by averaging the histograms for left and right sample trials). The z-scored firing rate was calculated based on the mean and standard deviation of the firing rate from 50 s before to 50 s after the event.

To statistically test the presence of sequences in firing activity, we use the ridge-to-background ratio, an approach previously introduced by (*Harvey et al., 2012*). For each neuron, we first calculate the average firing rates for each time bin relative to delay onset (using 0.1 s time bins). Ridge activity is the mean firing rate of the 11 bins (a 1.1 s-wide window) centered at the peak activity time. Background activity is defined to be mean firing rate of all the other time bins. We used a large window for calculating background activity, i.e. from 50 s before to 50 s after the aligning point. In *Figure 4—figure supplement 1B* those time bins that were masked were not included in the background activity calculation or ridge activity calculation. We tested the statistical significance with a one tailed test comparing the ridge-to-background ratio against the null distribution of ratios expected by chance. The null distribution was calculated by repeatedly circularly shifting spike times by random values.

In *Figures 4B,C*, and *Figure 4—figure supplement 2E*, firing activity peaks were selected from 1 s before the onset of the delay period to 1 s after the end of the delay period. In *Figure 4—figure supplement 2F* firing activity peaks were chosen from 5 s before to 15 s after the desired event. Likewise, the ridge-to-background analysis in *Figure 4—figure supplement 1C* was carried out on a time window spanning 5 s before to 15 s after the desired event. For all other heat-maps (*Figure 5B* and *Figure 4—figure supplement 2A*), scatter plots (*Figure 5C* and *Figure 4—figure supplement 2B*), and ridge-to-background ratio analyses (*Figure 4—figure supplement 1B*) firing activity peaks were chosen from the onset to the end of the delay period.

To calculate mutual information as described below, as well as to display time-varying firing rates for example neurons (*Figure 4A* and *Figure 5A*), the firing rates (calculated with bins of 50ms) were smoothed by convolving with a Gaussian kernel ($\sigma = 1/6$ s).

## Mutual information

Information about the sample stimulus identity in neural data was quantified by calculating the mutual information between spike trains and the sample side (*Borst and Theunissen, 1999*; *Dayan and Abbott, 2001*; *Quian Quiroga and Panzeri, 2009*; *Rieke et al., 1999*). We first made the two following simplifying assumptions: (1) the spiking of each neuron is a Poisson process (i.e. spiking at different time bins across neurons are independent of one-another) and, (2) the only factors involved in determining the spike rate is the sample stimulus of the current trial and time relative to the delay onset. Our calculations make use of the firing rates for left and right sample trials only, which are estimated as described above. The estimated firing rates are then used to generate 500 random spike trains. the average pointwise mutual information with the sample stimulus for each randomly generated spike train is calculated to obtain the mutual information. Spike trains are generated by assuming a sample stimulus x (randomly chosen to be left or right) and determining the number of spikes in every 50 ms time bin of the desired time segment from the estimated firing rates when <sample> = x. A 500 ms wide sequence $s = s_1, s_2, \ldots s_k$ of spike counts from the spike train is then analyzed. The pointwise mutual information between that sequence and the sample stimulus is derived as follows:

$$\log\left(\frac{Pr(<sample>=x \,\&\, <spike\ train>=s_1 s_2 \ldots s_k)}{Pr(<sample>=x)Pr(<spike\ train>=s_1 s_2 \ldots s_k)}\right)$$
$$= \log\left(\frac{Pr(<spike\ train>=s_1 s_2 \ldots s_k | <sample>=x)}{Pr(<spike\ train>=s_1 s_2 \ldots s_k)}\right)$$
$$= \log\left(\frac{Pr(<spike\ train>=s_1 s_2 \ldots s_k | <sample>=x)}{\sum_{L \in \{left;right\}} Pr(<spike\ train>=s_1 s_2 \ldots s_k | <sample>=L)}\right)$$

From independence between bins, we have:

$$Pr(<spike\ train>=s_1 s_2 \ldots s_k | <sample>=x)$$
$$= \prod_{i=1}^{k} Pr(<spikes\ in\ bin\ i>=s_i | <sample>=x)$$

And the term on the right can be calculated from the estimated firing rates in the case <sample> = x.

To determine if mutual information peaks are statistically significant, we compared mutual information peaks from the data to that obtained with shuffled data (*Panzeri et al., 2007*). Towards this end, we randomly shuffled the left sample and right sample trials and recalculated time-varying sample information using the approach explained above. For each neuron, this process was repeated 200 times and the maximum value of sample information within the delay period was taken each time, to obtain a distribution of the sample information peak value expected by chance. The peak information of the non-shuffled data was compared to that distribution to test for significance ($p<0.01$, one-tailed test). All neurons with peaks larger than the 99th percentile of the shuffled distribution were considered to be significantly encoding the sample stimulus (*Figure 5B*). The green shadings on the bottom panels of *Figure 5A* are the thresholds for significance of the peak value across the entire delay period interval, obtained by taking the 99th percentile of distribution for sample information peak expected by chance. Information about lever identity for choice and sample presses was calculated in a similar fashion, with the modification that we did not use a sliding window, but rather calculated for the entire 500 ms window centered at a lever press (*Figure 7C*).

## Population decoder

We used a maximum likelihood estimator to decode the sample stimulus from population data (*Brown et al., 2004*; *Gerwinn et al., 2009*; *Ma et al., 2006*; *Pouget et al., 2003*). Our decoder was constructed based on the assumption that spiking of each neuron is a Poisson process with a time-varying spike rate function. To obtain a time-dependent evaluation of the decoder, we use a 500 ms wide sliding window. For each neuron the desired 500 ms segment of the spike train is binned into 50 ms bins. The decoder is given an N×k matrix S, where $S_{i,j}$ is the spike count of neuron i for bin j of that segment (N denotes the number of neurons, and k is the number of bins). The decoder then finds x to maximize $Pr(<sample>=x \mid <spike\ data>=S)$. Assuming equal probability for left vs. right lever we have:

Assuming a uniform prior over the possible values for x, we have:

$$\Pr(<sample> = x \mid <spike\ data> = S) \propto \Pr(<spike\ train> = S \mid <sample> = x)$$

with the right term being the likelihood function. From independence between bins across neurons we know the likelihood function can be calculated as follows:

$$\Pr(<spike\ train> = S \mid <sample> = x)$$
$$= \prod_{i=1}^{N} \prod_{j=1}^{k} \Pr\big(<spikes\ in\ bin\ i\ of\ neuron\ j> = s_{i,j} \mid <sample> = x\big)$$

The term on the right can be calculated using the Poisson probability mass function and the spike rate of the corresponding bin under condition $<sample> = x$. We train the sample decoder by calculating time-varying firing rates for left sample and right sample trials separately, from the training data set. The sample decoder knows what time segment of the trial the spike train belongs to and uses firing rate estimates from that segment. Decoder is trained using correct trials only (i.e. trials in which the rat responded by choosing the correct lever).

Since the neurons were not all recorded simultaneously from the same rat, we randomly combined different trials from each neuron with one another into pseudo-trials. Trials were drawn with replacement, within a determined condition of the four trial conditions (i.e. [left sample, right sample] × [correct, incorrect]). 500 pseudo-trials were generated. The error bars in *Figure 6* are the SEM determined by the number of pseudo-trials if trials were drawn without replacement (i.e. the minimum number of trials for a specific condition across neurons), which is an upperbound on the SEM.

## Generalized linear models of spiking activity

Generalized linear models (GLMs) were constructed to predict the number of spikes of each neuron in each 10 ms time bin using various predictors related to the task and the animal's behavior. The GLMs modeled number of spikes with a Poisson distribution and a log link function). Positional predictors included: 9 binary predictors for head position, 8 binary predictors for head direction, and 3 numerical predictors for velocity (x component, y component, and magnitude of velocity vector). For the head position predictors, the chamber was divided into nine tiles (3 equal segments along each axis) and each binary predictor represented whether the head position was located in the corresponding tile. For the head direction predictors, the range of possible head directions (0–360 degrees) was divided into 8 equal segments and binary predictors for head direction represented which segment the head direction was in. For the task related predictors, the 10 binary predictors denoted time within the delay period and one binary predictor denoted the identity of the sample stimulus. For the time related predictors, time from delay onset was divided into 1 s long time bins and each of the 10 binary predictors signified whether the current time was in the corresponding 1 s time bin. Nested Model Comparison of GLMs were performed by first fitting the GLM to the data using R function glm() and then performing a likelihood ratio test using the R function anova (model1, model2, test = 'Chisq').

## Acknowledgements

We would like to thank J Pillow for helpful suggestions with the data analysis; C Brody and M Murthy for comments on this manuscript; and the rest of the Witten lab for feedback and help with this work. This work was supported by the Pew, McKnight, NARSAD and Sloan Foundations, NIH DP2 DA035149, NIH R01 MH106689, and an NSF GRFP.

## Additional information

### Funding

| Funder | Grant reference number | Author |
| --- | --- | --- |
| National Science Foundation | GRFP | Hessameddin Akhlaghpour |
| National Institute of Mental Health | 5R01MH106689-02 | Ilana B Witten |

| McKnight Foundation | | Ilana B Witten |
|---|---|---|
| Pew Charitable Trusts | | Ilana B Witten |
| National Institutes of Health | 1 DP2 DA035149-01 | Ilana B Witten |
| National Alliance for Research on Schizophrenia and Depression | | Ilana B Witten |
| Alfred P. Sloan Foundation | | Ilana B Witten |

The funders had no role in study design, data collection and interpretation, or the decision to submit the work for publication.

## Author contributions

HA, IBW, Conception and design, Acquisition of data, Analysis and interpretation of data, Drafting or revising the article; JW, Conception and design, Acquisition of data, Drafting or revising the article; JYC, JPT, JA, Acquisition of data, Analysis and interpretation of data, Drafting or revising the article

## Author ORCIDs

Joshua P Taliaferro, http://orcid.org/0000-0001-6051-8635
Ilana B Witten, http://orcid.org/0000-0003-0548-2160

## Ethics

Animal experimentation: This study was performed in strict accordance with the recommendations in the Guide for the Care and Use of Laboratory Animals of the National Institutes of Health. All of the animals were handled according to approved institutional animal care and use committee (IACUC) protocols (1876-15) of Princeton University. All surgery was performed under anesthesia, and every effort was made to minimize suffering.

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
