## [Decision Letter]

Thank you for submitting your article "Dissociated sequential activity and stimulus encoding in the dorsomedial striatum during spatial working memory" for consideration by *eLife*. Your article has been reviewed by three peer reviewers, and the evaluation has been overseen by Naoshige Uchida as the Reviewing Editor and Timothy Behrens as the Senior Editor. The following individual involved in review of your submission has agreed to reveal their identity: Earl Miller (Reviewer #2).

The reviewers have discussed the reviews with one another and the Reviewing Editor has drafted this decision to help you prepare a revised submission.

Summary:

The reviewers agreed that the authors have been very responsive to the reviewers' previous comments. In particular, the new inactivation experiment is important, and the authors added useful analyses to address the previous concerns. While the manuscript has improved significantly, the reviewers still raised some issues that need to be addressed before publication in *eLife*.

Essential points:

1) The decoding analysis shows that the neuronal activity contains sample information later in the delay (10 sec). Although the decoder performance is lower than that at the onset of delay period, the decoder performance is comparable to the behavioral performance (Figure 6). This would mean that the striatal activity at the end of delay period can provide sufficient information to support the animals' behavioral performance. This contradicts the authors' claims. We would like to see more robust discussions on this limitation in the conclusion in Discussion.

2) The authors discuss sequential activity throughout the manuscript but in some places, the existence of sequential activity is not very obvious (e.g. Figure 4—figure supplement 2). The correlation analysis that the authors performed might be problematic if correlations were drawn by a few outlier points. It would be helpful if the authors come up with a statistical way to quantify 'sequenceness' more directly. For instance, this may be achieved by looking at the rank in time of neurons recorded simultaneously across single trials.

3) In the inactivation experiment, it is important to report other behavioral measures to know whether DMS inactivation had other effects. Please report whether the range of doses used in the current experiment affected trials performed, time between trials, and other measures that can extracted.

4) The current task design may not be ideal for examining sample stimulus encoding during delay period. The authors have done a nice job of analyzing the data given what was done but some of the motor/place confounds can be more clearly excluded by the use of a more controlled task design. The authors now discuss their task design as having various advantages in Discussion but it would be useful to also point out disadvantages compared to other working memory paradigms commonly used in non-human primates and some rodent studies (e.g. block or randomly interleaved designs to dissociate sample stimulus from motor/place parameters). Please add some discussions on the limitation of the current task design with respect to strongly drawing conclusions on sample stimulus encoding.

[Editors’ note: a previous version of this study was rejected after peer review, but the authors submitted for reconsideration. The previous decision letter after peer review is shown below.]

Thank you for submitting your work entitled "Dissociated sequential activity and stimulus encoding in the dorsomedial striatum during spatial working memory" for consideration by *eLife*. Your article has been reviewed by three peer reviewers, and the evaluation has been overseen by Naoshige Uchida as the Reviewing Editor and Timothy Behrens as the Senior Editor. Our decision has been reached after consultation between the reviewers. Based on these discussions and the individual reviews below, we regret to inform you that your work will not be considered further for publication in *eLife*.

Previous studies have suggested that the striatum, in particular the dorsomedial striatum (DMS), may play a role in working memory. However, the mechanism by which the DMS is involved in working memory is not understood. The authors characterized firing patterns of DMS neurons in rats performing a delayed non-match to position (DNMP) task. The authors observed transient activation of neurons when the rats initiated a delay period by a nose poke. The authors also show that neurons were activated sequentially throughout the delay period. The main conclusion is that the activity at the onset of the delay period conveyed information about sample lever presses, the content of working memory, whereas the sequential activity did not. The authors suggest a possibility that the DMS is involved in working memory by gating information only at the early phase of working memory.

The reviewers find the main finding very interesting. However, during discussions as well as in the review comments, their enthusiasm differed significantly. Two reviewers were concerned about the possibility that the sequential activity as well as the transient activity at the beginning of the delay period are related to the movement or position of the animal and not directly related to working memory itself. The reviewers understood that the authors are not claiming that the sequential activation conveys information about working memory contents, but pointed out two problems. First, the data in Figure 6 shows that the information about sample lever presses are present throughout the delay period, so this contradicts with the analysis using mutual information. Second, what is causing the sequential activity remains unclear (i.e. is it really related to working memory, or other motor or place parameters?). The reviewers thought that the sequential activity is very interesting regardless of what it relates to. However, they thought that, to address these issues, more thorough analysis on behavior during the performance of the task is necessary although the task design appears to make it difficult to dissociate confounding factors completely. Another important issue is that it remains to be established whether the firing difference at the onset of the delay period is due to the content of working memory. The reviewers thought that showing that more information is available in neural activity when type of lever press (sample versus choice lever press) is specified is not sufficient to demonstrate this point.

*Reviewer #1:*

The study by Akhlaghpour and colleagues reports interesting data on ensemble activity in the striatum during a delayed non-matching to position task. The methods for data acquisition and analysis are more than adequate to address the target issues of the manuscript. However, several issues in the behavioral design reduced my enthusiasm for the paper, and really should be addressed before it is considered for publication.

First, can the authors show that perturbing sequential activity in the striatum alters the animals’ performance of the task in a way that is predicted by the striatum participating in working memory? You may not have data (yet) on this, but it would make the paper much stronger.

Second, the task that was used is somewhat limited in being able to find evidence for working memory given that only two options are presented to the animals. Despite this limitation, the study is still interesting and could provide novel insights into striatal function if two major issues are addressed. Major challenges for this research are (1) to dissociate the sequential activations that are reported from positional and locomotor behaviors that could be the true generators of the sequential activations and (2) to demonstrate that the sequential activations do not occur in a sensory-guided version of the task (for example, using visual cues). Resolving these issues are most important for establishing the validity of the authors' interpretations.

Also, how do the sequential activations relate to the persistent activity reported by Schultz and colleagues in early primate recordings in the striatum. They never referred to those activity patterns as working memory, but did claim to find such activations to be quite common within the striatum. Why do you think that you did not find such activities in your study? (I think that it is because your rats were freely moving and there were many behaviors that could influence the striatal neurons during the delays between responses.) I think that you should comment on this issue in the Discussion.

*Reviewer #2:*

This manuscript studies the relationship between sequential activity in the striatum and the code for spatial working memory using a DNMP task and electrical recordings. The paper is clearly written and clearly presented. The topic investigated is an interesting one. The analyses for the most part appear carefully performed and well done. I have one major concern relating to how well the authors can relate the activity they observed to working memory rather than other task variables, such as motor actions. This concern made it difficult for me to be convinced that the dissociation the authors propose, between sequential activity and working memory, is strongly supported. Without improvement in this area, I think the main conclusions cannot be stated with high confidence.

Detailed comments:

1) I am not convinced the authors have clearly shown that working memory activity is encoded in the population of striatal neurons recorded. The evidence for working memory activity is that there is mutual (MI) between the sample lever identity and firing rates during the delay period and that the decoder can read-out above chance levels the identity of the sample lever during the delay. However, it seems that another possibility is that the rat performs different actions following a left and right sample lever press and that the activity is related mostly to different motor actions on different trial types, rather than working memory. The encoding of motor actions might be reasonable for explaining why the MI is the highest at the start of the delay because the rat likely moves very differently away from the right and left levers. In addition, the decoder performance for sample lever identity is highest at the times during which the motor actions for the different sample lever trial types is most distinct, namely right after the sample lever press and during the response lever press (which involves different levers on correct trials for different samples). This point is consistent with the decoding of sample lever identity on error trials based on training the decoder on correct trials: the decoder performance is near perfect at the sample lever press and near zero at the response lever press. The one argument against this hypothesis is that presses of the same lever have different neural activity during sample and response periods. However, there are likely many different motor actions occurring in these cases, such as anticipatory licking during the response but not sample period or different body actions and orientations based on entrance into the port. Given the striatum's known role in driving motor actions, it seems that a perfectly valid interpretation of the data could be that the striatal firing is related to motor actions and is not related to working memory at all. With the current presentation of the behavioral data, it is impossible to understand if this alternative hypothesis can be excluded.

Therefore, I think more needs to be done to rule out the possibility that what the authors consider to be working memory signals are simply reflections of motor action differences of the rat on right and left sample lever trials. I acknowledge that this is a potentially never-ending source of additional analyses and measurements if all possible motor actions are considered. However, it seems like more could be done with the current data. Some straightforward plots and analyses could go a long way in helping the reader on this topic.

Do the rats have different movement patterns following right and left sample lever presses? The authors could plot movement trajectories at various time intervals after left and right sample lever presses and also at various time intervals preceding left and right response lever presses. These trajectories could be supported and summarized using heat maps for occupancy during left and right sample trials at various time points, as used currently in Figure 1. In these analyses, a marking of time and trajectories is critical, which is currently absent in Figure 1 heat maps.

In addition, the authors could use classifiers or decoding methods to determine if they can predict whether the sample lever pressed was left or right based on the rat's position, heading direction, movement speed, and all other behavioral parameters they have measured. This classification analysis could be performed as a function of time from sample lever press.

Can the authors decode movement-related signals from the striatal firing, such as current movement speed or upcoming actions (changes in heading direction)? If these signals exist, then it is possible that these signals are the dominant factors during the delay period. If the authors can decode things like current or upcoming movements from striatal firing, then they can ask if the types of decoded movements differ on left and right sample lever press trials.

These points are essential to the paper because if the authors want to say that working memory information is encoded and separable from the sequential activity, then it is critical to be very confident that the working memory signal is actually present in a form beyond motor actions.

2) The authors include only a subset of neurons for the analysis of MI in comparison to firing rate peaks. The inclusion criterion is whether at any point the MI goes above 99% of the shuffled MI values (p value of 0.01). I think the authors are doing many comparisons (many timepoints) per neuron for this MI significance test; however, they have not done a multiple comparisons correction. If the authors perform many comparisons in each neuron, then a multiple comparisons correction is needed, such as a Bonferroni correction. Without such a correction, a significant false positive rate, well above 1%, is expected.

*Reviewer #3:*

This is a straightforward and well-written manuscript describing neural dynamics in the striatum or rats performing a spatial working memory task. The task is a classic and the analyses and results convincing. The intriguing aspect of the results is that striatal neurons fire and different points in the delay such that peaks in spiking are spread throughout the memory delay. There is an interesting dissociation, however, peaks in information about the remembered sample occur at the beginning and end of the delay and thus does not line up with the peaks in firing.

I don't really have any major concerns. Just suggestions for improvement.

The authors rightly (as far as I know) point out that these results are unique in the rodent literature and likely made possible by the use of a 10 sec delay. However, there have been similar results reported in the primate cortex. For example, the peak of information at the beginning and end of the delay is seen quite commonly in the primate prefrontal and parietal cortex. I encourage the authors to look at this literature and cite some relevant studies.

Likewise, the sequential nature of activation in the delay has also been reported in primate prefrontal cortex, albeit for information. For example, check out Figure 4 of Cromer et al. 2011, J. Cog. Neuro. I think the authors should acknowledge and discuss this. And also, they need to be a little more cautious in their conclusion that the 10 sec delay was necessary. It may be in rodents, but not primates. It is interesting that primate cortex shows sequential information whereas rodent striatum less so. And interesting means worth mentioning.

Finally, there is an emerging class of working memory models that suggest that working memories are not encoded by sustained activity but instead by transient changes in synaptic weights induced by spiking activity. For example, consider Lunqvist et al. (2010, 2011, 2012) and the "activity-silent" models of Stokes. The result here seems to support this class of models seeing that activity isn't sustained across the delay. I think at this point should be raised.

---

## [Author Response]

*Summary:*

*The reviewers agreed that the authors have been very responsive to the reviewers' previous comments. In particular, the new inactivation experiment is important, and the authors added useful analyses to address the previous concerns. While the manuscript has improved significantly, the reviewers still raised some issues that need to be addressed before publication in eLife.*

*Essential points:*

*1) The decoding analysis shows that the neuronal activity contains sample information later in the delay (10 sec). Although the decoder performance is lower than that at the onset of delay period, the decoder performance is comparable to the behavioral performance (Figure 6). This would mean that the striatal activity at the end of delay period can provide sufficient information to support the animals' behavioral performance. This contradicts the authors' claims. We would like to see more robust discussions on this limitation in the conclusion in Discussion.*

Thanks for the clarification. We now point out in the Discussion that although much higher levels of information and decoding accuracy are available in DMS neural activity towards the onset of the delay period, the decoding accuracy remains above chance for much of the delay period, at a level that is roughly comparable to the behavioral performance. This means it is possible that information within DMS is used for maintaining the shortterm memory. The only way to definitively address the question of whether DMS is more important for updating versus maintaining the memory is through future experiments involving fast timescale, subtrial manipulations, which could potentially distinguish on a causal level between these two possibilities (see subsection “Relevance to theoretical models of the striatal contribution to working memory” last paragraph).

*2) The authors discuss sequential activity throughout the manuscript but in some places, the existence of sequential activity is not very obvious (e.g. Figure 4—figure supplement 2). The correlation analysis that the authors performed might be problematic if correlations were drawn by a few outlier points. It would be helpful if the authors come up with a statistical way to quantify 'sequenceness' more directly. For instance, this may be achieved by looking at the rank in time of neurons recorded simultaneously across single trials.*

As an additional measure to quantify the presence of sequences in the firing rates, we now include the ridgetobackground analysis, an approach previously established for this specific purpose by Harvey et al., 2012. The ridgetobackground analysis takes ridge activity (defined as average activity in a fixed time window around the peak firing rate time) divided by background activity for each neuron, and statistically compares this ratio in the real data to a null distribution which is obtained by repeatedly shifting spike times in the behavioral session. As expected, we found that the ridgetobackground ratio was significantly above chance levels across all conditions presented in the paper (see Figure 4—figure supplement 1).

*3) In the inactivation experiment, it is important to report other behavioral measures to know whether DMS inactivation had other effects. Please report whether the range of doses used in the current experiment affected trials performed, time between trials, and other measures that can extracted.*

To supplement the accuracy and latency measures reported in Figure 2, we now report the effect of muscimol on the number of trials performed, sample omission rates, trials abort rates, sample bias, and choice bias (see Figure 2—figure supplement 1). Unlike accuracy, none of these other measures were significantly affected by muscimol across the population of rats (n=7). However, a few of the rats did seem to perform fewer trials with DMS inactivation (with more sample omissions and more trial aborts), which is consistent with DMS being implicated in motivation or response vigor (Wang et al. 2013). Even if this effect were to become statistically significant across a larger population of rats, our conclusion stands that silencing of DMS activity impairs accuracy in the delayed nonmatch to sample task for trials that are completed.

*4) The current task design may not be ideal for examining sample stimulus encoding during delay period. The authors have done a nice job of analyzing the data given what was done but some of the motor/place confounds can be more clearly excluded by the use of a more controlled task design. The authors now discuss their task design as having various advantages in Discussion but it would be useful to also point out disadvantages compared to other working memory paradigms commonly used in non-human primates and some rodent studies (e.g. block or randomly interleaved designs to dissociate sample stimulus from motor/place parameters). Please add some discussions on the limitation of the current task design with respect to strongly drawing conclusions on sample stimulus encoding.*

Thank you for noting this. We have added a paragraph to the Discussion on the limitations of our task design. Motor confounds may have been more strongly excluded had we used a task design that restricts the movement of the rat during the delay period. Additionally, our task makes it difficult to distinguish between motor planning and sample stimulus memory because the correct choice lever can be determined from the beginning of the delay period (see subsection “Insights into the striatal dynamics underlying working memory” third paragraph).

[Editors’ note: the author responses to the previous round of peer review follow.]

*[…] The reviewers find the main finding very interesting. However, during discussions as well as in the review comments, their enthusiasm differed significantly. Two reviewers were concerned about the possibility that the sequential activity as well as the transient activity at the beginning of the delay period are related to the movement or position of the animal and not directly related to working memory itself. The reviewers understood that the authors are not claiming that the sequential activation conveys information about working memory contents, but pointed out two problems. First, the data in Figure 6 shows that the information about sample lever presses are present throughout the delay period, so this contradicts with the analysis using mutual information.*

We appreciate the thorough attention to the details of our results. Regarding the potential discrepancy between Figure 5 and Figure 7 (previously 4B and 6A), the two panels were generated using two different methods to quantify the encoding of the sample stimulus during the delay period. Our first method (Figure 5) was to quantify the mutual information between the sample stimulus and the firing patterns for each individual neuron, only including the units with significant sample information peaks. That method revealed strong encoding at the onset of the delay period and very little encoding during the delay period (only a few neurons appear to encode the memory of the sample stimulus above 0.2 bits later in the delay period). Our second method (Figure 7) was to measure our ability to decode the sample stimulus from the entire neural population as a function of time during the delay period. The second method revealed perfect decoding accuracy at the onset of the delay period and greater than chance level accuracy during most of the delay period. Based on the Reviewer comment, we now appreciate that these two results may at first appear to contradict each other, since a small amount of sample information in a few neurons contributes to an abovechance level of sample decoding accuracy when using the population data. However, the takehome message from the two plots is the same: the sample stimulus is encoded strongly at the onset of the delay period and that sample encoding declines noticeably with either measure by 23s into the delay period. We briefly clarify this in the Discussion of the revised manuscript (subsection “Insights into the striatal dynamics underlying working memory”, first paragraph).

*Second, what is causing the sequential activity remains unclear (i.e. is it really related to working memory, or other motor or place parameters?). The reviewers thought that the sequential activity is very interesting regardless of what it relates to. However, they thought that, to address these issues, more thorough analysis on behavior during the performance of the task is necessary although the task design appears to make it difficult to dissociate confounding factors completely.*

Thank you for commenting that the sequential activity is very interesting, and for raising an important issue that needs to be addressed. To demonstrate that the sequences are not a byproduct of movement or position, we use the head tracking data and show that even when the rat’s head is in the nose port and therefore position is controlled for, the sequences are still present. This analysis is described in detail in our pointbypoint responses to Reviewer #1 below.

*Another important issue is that it remains to be established whether the firing difference at the onset of the delay period is due to the content of working memory. The reviewers thought that showing that more information is available in neural activity when type of lever press (sample versus choice lever press) is specified is not sufficient to demonstrate this point.*

We believe you have raised an important question regarding whether the sample encoding we observe at the onset of the delay period is in fact simply encoding of position or motor actions. To support our claim that the firing rate differences observed between left and right sample presses cannot be accounted for merely by position, motion, or motor actions, we compare neural correlates of the sample press with neural correlates of the choice press (Figure 7). In our revised submission, we do so by examining a much shorter time window around the time of the lever press (only ± 0.25s rather than ± 1s as in our original submission). This narrow time window was chosen based on examination of the head tracking data to minimize movement towards and away from the lever. Our analysis reveals that DMS neurons encode choice presses and sample presses differently. We provide details regarding this analysis in our responses to Reviewer #2 below.

In addition, we now provide new analysis of sample encoding when the rats first arrived at the nose port, shortly after pressing the sample lever. In this case, we show that head position is maintained at a similar location for left and right sample trials. Yet we observe significant sample encoding (Figure 6—figure supplement 1). This further supports our claim that sample encoding during the delay period cannot be explained by the rats’ position and motion.

*Reviewer #1:*

*The study by Akhlaghpour and colleagues reports interesting data on ensemble activity in the striatum during a delayed non-matching to position task. The methods for data acquisition and analysis are more than adequate to address the target issues of the manuscript. However, several issues in the behavioral design reduced my enthusiasm for the paper, and really should be addressed before it is considered for publication.*

Thank you very much for describing our data as interesting and for mentioning that the data acquisition and analysis are more than adequate. As you will see below, we have worked to address your concern about the influence of the subject’s movement and position on our conclusions through additional analyses and experiments.

*First, can the authors show that perturbing sequential activity in the striatum alters the animals’ performance of the task in a way that is predicted by the striatum participating in working memory? You may not have data (yet) on this, but it would make the paper much stronger.*

Thank you for the suggestion. We agree that this is a good experiment which would make the paper much stronger. To this end, we trained a new cohort of rats on the delayed nonmatch to position task and inactivated the DMS with bilateral infusion of muscimol. Infusion of muscimol in the DMS led to much lower accuracy when compared to infusion of saline (Figure 2, p < 1e6 repeated measures ANOVA, n=7 rats). The fact that rats press the incorrect choice lever more frequently when the DMS is inactivated suggests that the DMS activity contributes to spatial working memory in this task.

*Second, the task that was used is somewhat limited in being able to find evidence for working memory given that only two options are presented to the animals. Despite this limitation, the study is still interesting and could provide novel insights into striatal function if two major issues are addressed. Major challenges for this research are (1) to dissociate the sequential activations that are reported from positional and locomotor behaviors that could be the true generators of the sequential activations and (2) to demonstrate that the sequential activations do not occur in a sensory-guided version of the task (for example, using visual cues). Resolving these issues are most important for establishing the validity of the authors' interpretations.*

We very much appreciate the suggestion to better account for positional variables in our analyses. We are confident that our paper is much stronger as a result of the analytical approaches that we describe below.

To demonstrate that sequential activations cannot be explained by positional or locomotor behavior, we took two distinct and complementary approaches.

In our first approach, we plotted the activity of DMS neurons only for the subset of time that the rat was at the nose port, effectively controlling for body position and locomotion. The rats spend the majority of the delay period positioned at the nose port, waiting for the end of the delay period (in Figure 1—figure supplement 1).

We recalculated the firing rate sequences using only the subset of time during which the rats were positioned at the nose port, thereby controlling for position (Figure —figure supplement 1AD). The firing rates were calculating by aligning spiking activity to delay period onset (Figure 4—figure supplement 2, leftmost panel) or to time of arrival at the nose port (Figure 4—figure supplement 2, leftmost panel). Time bins in which the rat was not at the nose port for a sufficient amount of time to calculate firing rates were excluded (masked in white in Figure 4—figure supplement 2 & Figure 4—figure supplement 2; see Methods). As you can see below, there is a high degree of correlation in the time of peak firing calculated by only using data from when the animal was at the nose port versus using the entire dataset (Figure 4—figure supplement 2 for alignment to noseport arrival, r = 0.72, p < 106 Pearson correlation test; and Figure 4—figure supplement 2 for alignment to delay period onset, r = 0.36, p < 0.05 Pearson correlation test). This demonstrates that sequences were not simply neural correlates of position or locomotion, and occurred even when the rats remained at the nose port.

In our second approach to demonstrate that sequential firing activity cannot be explained by position and locomotion, we used a statistical model to show that time in the delay period is a significant predictor of neural activity, even when taking into account the rats’ position/locomotor variables. What this means is that temporal fluctuations in firing rate that occurs during the delay period are not merely a byproduct of position/locomotion variables. More specifically, we modeled each neuron’s spiking activity using two generalized linear models (GLMs), one with position related predictors only (i.e. head position, head direction, and velocity – all calculated using head tracking) and the other with positionrelated predictors in conjunction with time from the delay period onset as a predictor. We demonstrate that time from delay onset significantly improves the model in 80% of the neurons (84/105) in comparison to a model with only the position/locomotor variables (p < 0.0001, likelihood ratio test comparing models with and without time from delay period as predictor). This demonstrates that the timedependent changes in firing rates during the delay period cannot be accounted for by position or the other variables we tested. (see Methods for more details on the model).

Thank you for mentioning that it would be interesting to know if the sequences would occur in a sensory version of the task. Although we do not have this data, we do want to point out that our conclusions do not rest on the assumption that sequential activity would not appear in a sensory guided version of the task. In fact, we think it is likely that we would indeed observe sequential activity in a guided version of the task, given that we see sequential activation of neurons even outside the delay period (see Figure 4—figure supplement 2). Furthermore, the presence of sequences in a sensory guided version of the task will not necessarily be informative as to whether these sequences are related to working memory. It could be that sequential activation enables working memory, but is nevertheless present in tasks that are not related to working memory.

*Also, how do the sequential activations relate to the persistent activity reported by Schultz and colleagues in early primate recordings in the striatum. They never referred to those activity patterns as working memory, but did claim to find such activations to be quite common within the striatum. Why do you think that you did not find such activities in your study? (I think that it is because your rats were freely moving and there were many behaviors that could influence the striatal neurons during the delays between responses.) I think that you should comment on this issue in the Discussion.*

Thank you for raising this question. In the Discussion section of our new submission (subsection “Insights into the striatal dynamics underlying working memory”, second paragraph) we now elaborate on how our study relates to previous studies in primates. Previous recordings in primates show sustained delay period activity that encodes future choice in a substantial fraction of caudate neurons (Schultz et al. 1994, Hikosaka et al. 1989, Schultz & Romo 1988; Kawagoe et al., 1998). However the delay periods in these studies were typically 24s, which is much shorter than the 10s delays used in our task, and there was a low proportion of neurons that showed sustained activity over that delay period (less than 10% of task modulated caudate neurons had sustained activity in one study that quantified this with delay periods up to 4s, i.e. Hikosaka et a. 1989, and less than a third of recorded neurons were found to have sustained memory related activity in another study that used delay periods 11.5s long, i.e. Kawagoe et al. 1988). In our study we do observe some neurons with sample encoding during the delay period that persisted for 24s (such as the one illustrated in Figure 8), in agreement with prior work. Given that the prior studies did not employ longer delay periods, we do not think our new data contradicts the prior studies.

Author response image 1.**DOI:**
http://dx.doi.org/10.7554/eLife.19507.014

*Reviewer #2:*

*This manuscript studies the relationship between sequential activity in the striatum and the code for spatial working memory using a DNMP task and electrical recordings. The paper is clearly written and clearly presented. The topic investigated is an interesting one. The analyses for the most part appear carefully performed and well done. I have one major concern relating to how well the authors can relate the activity they observed to working memory rather than other task variables, such as motor actions. This concern made it difficult for me to be convinced that the dissociation the authors propose, between sequential activity and working memory, is strongly supported. Without improvement in this area, I think the main conclusions cannot be stated with high confidence.*

Thank you very much for the positive comments about our manuscript and the analysis. Below, we have performed new analyses to address your concerns, based on using the head tracking data collected during the recordings.

*Detailed comments:*

*1) I am not convinced the authors have clearly shown that working memory activity is encoded in the population of striatal neurons recorded. The evidence for working memory activity is that there is mutual (MI) between the sample lever identity and firing rates during the delay period and that the decoder can read-out above chance levels the identity of the sample lever during the delay. However, it seems that another possibility is that the rat performs different actions following a left and right sample lever press and that the activity is related mostly to different motor actions on different trial types, rather than working memory. The encoding of motor actions might be reasonable for explaining why the MI is the highest at the start of the delay because the rat likely moves very differently away from the right and left levers. In addition, the decoder performance for sample lever identity is highest at the times during which the motor actions for the different sample lever trial types is most distinct, namely right after the sample lever press and during the response lever press (which involves different levers on correct trials for different samples). This point is consistent with the decoding of sample lever identity on error trials based on training the decoder on correct trials: the decoder performance is near perfect at the sample lever press and near zero at the response lever press. The one argument against this hypothesis is that presses of the same lever have different neural activity during sample and response periods. However, there are likely many different motor actions occurring in these cases, such as anticipatory licking during the response but not sample period or different body actions and orientations based on entrance into the port. Given the striatum's known role in driving motor actions, it seems that a perfectly valid interpretation of the data could be that the striatal firing is related to motor actions and is not related to working memory at all. With the current presentation of the behavioral data, it is impossible to understand if this alternative hypothesis can be excluded.*

*Therefore, I think more needs to be done to rule out the possibility that what the authors consider to be working memory signals are simply reflections of motor action differences of the rat on right and left sample lever trials. I acknowledge that this is a potentially never-ending source of additional analyses and measurements if all possible motor actions are considered. However, it seems like more could be done with the current data. Some straightforward plots and analyses could go a long way in helping the reader on this topic.*

*Do the rats have different movement patterns following right and left sample lever presses? The authors could plot movement trajectories at various time intervals after left and right sample lever presses and also at various time intervals preceding left and right response lever presses. These trajectories could be supported and summarized using heat maps for occupancy during left and right sample trials at various time points, as used currently in Figure 1. In these analyses, a marking of time and trajectories is critical, which is currently absent in Figure 1 heat maps.*

Thank you for these helpful comments and suggestions. As you mention, our claim is that we see encoding of the sample stimulus towards the onset of the delay period and that this sample encoding is not simply a byproduct of encoding of body position, location, or motor actions. We support this claim by the following approaches: first by more carefully examining the onset of the delay period when the rat presses the sample lever, and second by examining the time during the delay period during which the rats’ position for the right and left sample trials converge.

Before describing the details of these analyses, we want to point out that although we are controlling for the rats’ head position and action in both of these analyses, we recognize that what we refer to as sample encoding may in fact be encoding of an action intended to be executed in the future, sometimes referred to as premotor activity or prospective memory. In fact, given that there is a one-to-one mapping between the identity of the sample lever and the correct choice lever to be pressed at the offset of the delay period, our task design does not allow us to easily distinguish between retrospective and prospective memory (i.e. sample encoding versus premotor activity). We now bring up this important caveat up in the Discussion- thank you for bringing it up. It is worth noting that in the second analysis described below for which we examine the time during the delay period during which the head position trajectories converge, the head position is similar for left and right trials for several seconds following the time window being analyzed, as will be evident in the occupancy maps.

In the first analysis, we show that the firing patterns at the onset of the delay period cannot be fully accounted for by position or action, by showing dramatic differences in encoding of lever identity for sample presses versus choice presses. This is a meaningful comparison because the rats are doing similar actions (i.e. pressing the left or right lever), but in a different context within the trial (i.e. sample vs choice press). We have now replotted the sample versus choice analysis in Figure 7 (previously Figure 5) using a shorter time window (only ± 0.25s around the lever press, rather than ± 1s as before). The new time window was chosen to minimize movement before and after the lever press (see similarity in head position for sample and choice lever presses during this time window in 7A). This ensures that we do not include any additional motor actions such as anticipatory licking at the reward port, or movement towards or away from the lever for this analysis. In Figure 7, we report the similarity in head position of the choice lever presses and sample lever presses. Next, in Figure 7, we show an example neuron that responds to right sample presses but not to right choice presses. This observation cannot be explained merely as a result of encoding of motor actions, location, or body position since the rat performs the same action at a similar location but we observed very different neural responses. Much like the example neuron (Figure 7), across the population of recorded neurons (Figure 7), some neurons preferentially encoded lever identity during the sample press, while others during the choice press. In fact, there was significantly more dispersion in the difference between sample and choice information than expected by chance (inset in Figure 7 < 105; two sample Ftest for equality of variance for (sample information choice information) in real vs. shuffled data). This analysis supports the conclusion that DMS firing patterns are not merely reflecting position or motor actions.

In our second analysis, we instead examine a time point during the delay period during which the movement trajectories converge for left and right sample trials. Following the sample press, rats move to the opposite wall and wait in front of the nose port for the delay period to end (Figure 1—figure supplement 1). For this analysis, instead of aligning trials to the beginning of the delay period (i.e. the sample lever press, Figure 6), we align trials to the moment the rats first arrive at the nose port, excluding the small subset of trials in which they left the nose port within 1.5 seconds of the nose port arrival. Note that the head position is very similar for left and right sample trials at the nose port arrival time and in the following seconds (Figure 6—figure supplement 1). We then assess the ability of our population decoder to decode the sample stimulus from the neural data using leaveoneout cross validation with a 0.5s wide sliding window (same approach used in Figure 6). We noted high decoding accuracy, even during the first 1.5s during which the rat always maintains its head position in front of the nose port (see Figure 6—figure supplement 1). This is important as it provides evidence of memory encoding within striatal neurons during the delay period while carefully controlling for the rats’ position on left vs right sample trials. Note that the decoding accuracy is highest preceding the nosepoke arrival, which is consistent with our conclusion that sample encoding is highest at the onset of the delay period.

*In addition, the authors could use classifiers or decoding methods to determine if they can predict whether the sample lever pressed was left or right based on the rat's position, heading direction, movement speed, and all other behavioral parameters they have measured. This classification analysis could be performed as a function of time from sample lever press.*

*Can the authors decode movement-related signals from the striatal firing, such as current movement speed or upcoming actions (changes in heading direction)? If these signals exist, then it is possible that these signals are the dominant factors during the delay period. If the authors can decode things like current or upcoming movements from striatal firing, then they can ask if the types of decoded movements differ on left and right sample lever press trials.*

*These points are essential to the paper because if the authors want to say that working memory information is encoded and separable from the sequential activity, then it is critical to be very confident that the working memory signal is actually present in a form beyond motor actions.*

These are helpful suggestions to address the possibility of sample selectivity and sequential activity being a byproduct of position/motor variables such as head position, head direction, and velocity. In fact, we feel it is reasonable to expect neurons to display a mixture of selectivity for position/motor variables and for memories of taskrelated variables. However, one of our conclusions is that neural activity cannot be fully explained without taskrelated variables, such as the memory of sample stimulus or time from delay onset. To determine if this conclusion is justified, we sought to determine quantitatively if taskrelated variables had predictive power, even when accounting for motor variables. Towards this end, we modeled neural activity based on both task related variables and motor related variables. More specifically, we modeled each neuron's spiking activity using two generalized linear models (GLMs): one with only position related predictors and the other with both taskrelated and positionrelated predictors. The position/motion predictors we used were head position, head direction, and head velocity and the taskrelated variables were time from delay period onset, sample stimulus of current trial, and the interaction between time and sample stimulus. We performed a nested model likelihood ratio test for each neuron to compare the two models, to determine if the addition of the taskrelated variables improved the model (note that a likelihood ratio test controls appropriately for the difference in number of parameters between the models and that these models were based on delayperiod data only). For more than 90% of neurons, the model significantly improved when including the taskrelated variables (p < 0.0001, likelihood ratio test comparing models with and without task related variables as predictor). This provides quantitative evidence that there is significant taskrelated encoding in these neurons, even when quantitatively accounting for the motor variables.

Again, thank you for the suggestions to better account for head position and movements in our analyses – we are confident that our paper is much stronger as a result.

*2) The authors include only a subset of neurons for the analysis of MI in comparison to firing rate peaks. The inclusion criterion is whether at any point the MI goes above 99% of the shuffled MI values (p value of 0.01). I think the authors are doing many comparisons (many timepoints) per neuron for this MI significance test; however, they have not done a multiple comparisons correction. If the authors perform many comparisons in each neuron, then a multiple comparisons correction is needed, such as a Bonferroni correction. Without such a correction, a significant false positive rate, well above 1%, is expected.*

Thank you for this comment. In fact, we do one comparison for each neuron – not one comparison for each timepoint. The threshold is defined as the expected peak value across the entire time period, regardless of when the peak information occurs. With the current method, we expect the number of false positive neurons to be 1%. We appreciate that you pointed this out as a potential source of ambiguity and so we now provide a clearer explanation in the Methods.

*Reviewer #3:*

*This is a straightforward and well-written manuscript describing neural dynamics in the striatum or rats performing a spatial working memory task. The task is a classic and the analyses and results convincing. The intriguing aspect of the results is that striatal neurons fire and different points in the delay such that peaks in spiking are spread throughout the memory delay. There is an interesting dissociation, however, peaks in information about the remembered sample occur at the beginning and end of the delay and thus does not line up with the peaks in firing.*

Thank you for your positive comments. We are delighted that you found our study intriguing and that you share our enthusiasm about these findings.

*I don't really have any major concerns. Just suggestions for improvement.*

*The authors rightly (as far as I know) point out that these results are unique in the rodent literature and likely made possible by the use of a 10 sec delay. However, there have been similar results reported in the primate cortex. For example, the peak of information at the beginning and end of the delay is seen quite commonly in the primate prefrontal and parietal cortex. I encourage the authors to look at this literature and cite some relevant studies.*

Thank you for these points. We now cite some classic papers and also some more recent studies that report stronger activity and more information at the onset of the delay period in LIP and FEF (Clark, Noudoost, and Moore 2012; Bruce et al. 1985; Colby, Duhamel, and Goldberg 1996; Gregoriou, Gotts, and Desimone 2012).

*Likewise, the sequential nature of activation in the delay has also been reported in primate prefrontal cortex, albeit for information. For example, check out Figure 4 of Cromer et al. 2011, J. Cog. Neuro. I think the authors should acknowledge and discuss this. And also, they need to be a little more cautious in their conclusion that the 10 sec delay was necessary. It may be in rodents, but not primates. It is interesting that primate cortex shows sequential information whereas rodent striatum less so. And interesting means worth mentioning.*

Thank you for the additional reference we now cite (Cromer et al. 2011). In addition, we agree that is a good point that there may be important species differences in neural correlates of working memory between rodents and primates. We now mention this in the Discussion.

*Finally, there is an emerging class of working memory models that suggest that working memories are not encoded by sustained activity but instead by transient changes in synaptic weights induced by spiking activity. For example, consider Lunqvist et al. (2010, 2011, 2012) and the "activity-silent" models of Stokes. The result here seems to support this class of models seeing that activity isn't sustained across the delay. I think at this point should be raised.*

This is a very good point thank you for bringing this up. We now reference the models of (Mongillo, Barak, and Tsodyks 2008; Stokes 2015; Lundqvist et al. 2010) in our Discussion as a possible interpretation of our results.